# AgenticMath: Enhancing LLM Reasoning via Agentic-based Math Data Generation

## Abstract

The creation of high-quality datasets to improve Large Language Model (LLM) reasoning remains a significant challenge, as current methods often suffer from generating low-quality/incorrect answers and limited information richness from available data sources. To address this, we propose **AgenticMath**, a novel agentic pipeline for generating high-quality mathematical question-answer pairs to enhance the supervised fine-tuning of LLMs. Our method operates through four stages: (1) *Seed Question Filter* that selects questions with high information richness, complexity, and clarity; (2) an *Agentic Question Rephrase* step that employs a multi-agent system to generate diverse, logically consistent paraphrases; (3) an *Answer Augment* step where rewrite answers using chain-of-thought reasoning to enhance numerical and logical correctness, without reliance on human-provided labels; and (4) a final *Question and Answer Evaluation* that retains only the most superior pairs. Extensive experiments demonstrate that, fine-tuning 3B-8B parameter LLMs on **AgenticMath** generated datasets (comprising only 30-60K math samples) achieves competitive or superior performance on diverse in domain and out-of-domain mathematical reasoning benchmarks compared to baselines trained on much more data (e.g., 400K or 2.3M samples). Our work demonstrates that targeted, high-quality data generation is a more efficient path to improving mathematical reasoning in LLMs than large-scale, low-quality alternatives.

## 1 Introduction

Large language models (LLMs) (Brown et al., 2020; Achiam et al., 2023; Chowdhery et al., 2023; Touvron et al., 2023) have achieved strong results across many domains, showing impressive general reasoning and knowledge transfer. However, when applied to mathematical reasoning, open models (Touvron et al., 2023; Bai et al., 2023; Bi et al., 2024) still perform far below human levels, struggling with the precision and consistency required. Mathematical problems demand long chains of logic that combine symbolic manipulation, cross-domain knowledge, and step-by-step numerical accuracy (Ahn et al., 2024; Long et al., 2024). These requirements make math reasoning more complex and error-prone than typical natural language tasks.

**Limitations in Existing Math Reasoning Methods.** To improve the mathematical proficiency of LLMs, research has mainly followed two paths. The first uses prompt engineering (Fu et al., 2022), such as Chain-of-Thought (Wei et al., 2022) and Self-Consistency (Wang et al., 2022), which guide models to produce reasoning chains at test time. This method is simple and training-free but its gains are limited by model capacity and often unstable across problem types. The second path relies on powerful base models to synthesize large numbers of question–solution pairs for supervised fine-tuning (SFT) (Yu et al., 2023; Li et al., 2024a; Yue et al., 2023; Gou et al., 2023). This reduces annotation costs and boosts benchmark scores, yet performance is capped by the quality of the synthetic data. When generated problems lack clarity, rigor, or diversity, the resulting models remain far below the performance attainable with expert-annotated corpora. The core challenge is not just producing solutions but enforcing strict quality control during problem synthesis, since the problem statement shapes both the reasoning process and the useful information in the dataset.

**Limitations in Multi-Agent Data Generation for Mathematics.** Recent work has introduced LLM-based multi-agent frameworks to improve synthetic corpora (Abdullin et al., 2024; Chen et al.,

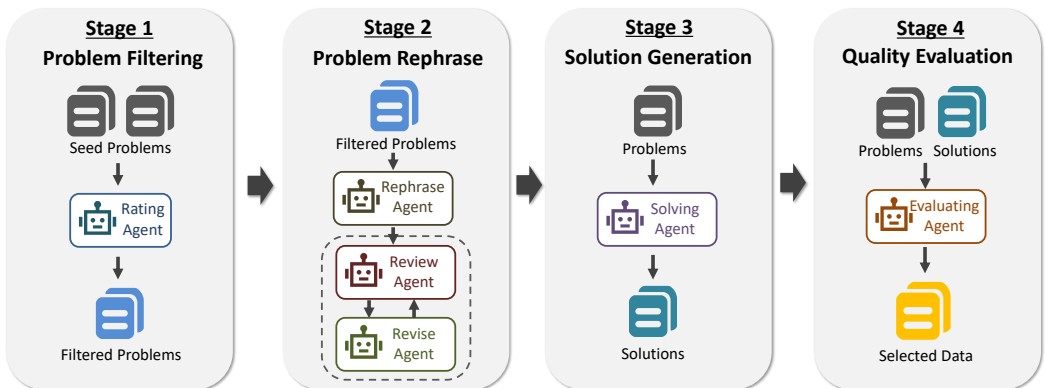

Figure 1: The Overview of AgenticMath Pipeline.

2024a; Mitra et al., 2024a; Ge et al., 2024; Chen et al., 2024b; Ye et al., 2024). Most of these methods target general-purpose instruction data, where task formulation is relatively shallow. In mathematics, the quality of the problem itself is decisive: precise formulation, logical coherence, and sufficient variability not only ensure solvability but also drive the generation of rigorous solutions. Without careful problem design, even advanced solution-generation strategies cannot compensate for poorly posed questions, keeping the dataset far from its upper bound. Existing multi-agent methods rarely enforce such domain-specific constraints, and prior attempts in mathematical data generation (Mitra et al., 2024b; Motwani et al., 2024) still lack systematic control at the problem construction stage.

**How AgenticMath Tackles the Challenges.** To address these challenges, we propose *Agentic-Math*, an automated multi-agent framework that enforces quality control at every stage of mathematical data generation. The framework leverages LLMs for generation, evaluation, and coordinated decision-making. It proceeds in four stages: (1) *Seed filtering* extracts high-value problems from human-authored corpora; (2) *Problem synthesis* engages cooperative agents to rephrase and diversify questions under explicit quality-control criteria; (3) *Solution generation* employs a solver agent to produce complete reasoning chains with rigor and correctness; and (4) *Quality evaluation* aggregates multi-dimensional scores to assess each problem–solution pair. By retaining only top-scoring samples, AgenticMath resolves the data quality bottleneck and follows the *"Less Is More"* principle. The result is a **data-efficient, high-quality dataset** that directly addresses the challenges of clarity, rigor, and diversity in mathematical reasoning tasks.

**Empirical Results and Contributions.** We evaluate AgenticMath on six mathematical reasoning benchmarks, including in-domain tasks (GSM8K (Cobbe et al., 2021), MATH (Hendrycks et al., 2021)) and out-of-domain settings (CollegeMath (Tang et al., 2024), DeepMind Mathematics (Saxton et al., 2019), OlympiadBench (He et al., 2024), TheoremQA (Chen et al., 2023)). AgenticMath matches or surpasses previous methods that rely on hundreds of thousands or even millions of samples (e.g., 400K or 2.3M), while using far fewer data. With **only 30K–60K** samples, performance improves by over 10 points on average, showing clear data efficiency and strong generalization to out-of-domain tasks. These results establish *AgenticMath* as an efficient and competitive approach to advancing mathematical reasoning. The main contributions of this work are as follows:

• **Agentic Math Data Generation:** We propose *AgenticMath*, an effective multi-agent framework for synthesizing, evaluating, and refining mathematical problems and solutions, offering a systematic and scalable paradigm for building high-quality reasoning corpora.

• **High-Quality Math Data:** We release *AgenticMathQA*, a curated dataset in 30K, 60K, and 90K versions. Unlike prior approaches that rely on scale, our dataset emphasizes clarity, correctness, and diversity, providing higher-quality supervision for mathematical reasoning.

• **Comprehensive Empirical Validation and Insights:** Extensive experiments show that with **only 5%–15%** of the data size, *AgenticMath* matches or even surpasses methods trained on 400K–2M samples. This result demonstrates that data quality, rather than dataset scaling alone, is the main factor behind improvements in mathematical reasoning.

## 2   RELATED WORK

**LLM for Math Reasoning.**   Large language models (Brown et al., 2020; Achiam et al., 2023; Touvron et al., 2023; **?**; Chowdhery et al., 2023; Bi et al., 2024; Team et al., 2023; 2024; Grattafiori et al., 2024) show strong general ability and are increasingly applied to mathematical problem solving (Cobbe et al., 2021; Hendrycks et al., 2021; Zhang et al., 2024a; Xia et al., 2025). Prompt-based approaches (Wei et al., 2022; Wang et al., 2022; Fu et al., 2022) extend reasoning paths but yield limited improvements. Recent work thus emphasizes synthesizing math reasoning data for supervised fine-tuning (Yu et al., 2023; Luo et al., 2023; Tang et al., 2024; Li et al., 2024a; Zhang et al., 2024b; Liu et al., 2025a; Tong et al., 2024). WizardMath (Luo et al., 2023) adds evolution directives and reinforcement learning; MathFusion (Pei et al., 2025) fuses problems for relational reasoning. Other methods integrate code tools (Yue et al., 2023; Wang et al., 2023; Hosseini et al., 2014; Toshniwal et al., 2024; Li et al., 2024b; Lu et al., 2024). In this work, we advance mathematical reasoning by improving both the question formulation and answer quality in synthetic data.

**Multi-Agent for Data Generation**   Multi-agent systems (Hong et al., 2023; Wu et al., 2023; Li et al., 2023; aut, 2023) show strong ability and are increasingly applied to data synthesis. Abdullin et al. (2024) proposed a multi-intelligence framework for dialog generation, while MAGDi (Chen et al., 2024a) used graph-based interactions and MALLM-GAN (Ling et al., 2024) employed generator–discriminator agents for tabular data. AgentInstruct (Mitra et al., 2024a) and Orca-Math (Mitra et al., 2024b) iteratively refined instructions, whereas role-driven synthesis was explored by Ge et al. (2024) and VCR (Liu et al., 2025b). MALT (Motwani et al., 2024) introduced generator, verifier, and refiner agents for math problems. Despite these advances, ensuring high-quality data for mathematical reasoning remains challenging. To address this, we introduce seed filtering and quality evaluate agents to guarantee reliable math reasoning data.

## 3   AGENTICMATH: MULTI-AGENT DESIGN FOR MATH REASONING

This section details the proposed AgenticMath (see Figure 1), which is designed to generate high-quality mathematical problems and reasoning solutions based on the GSM8K (Cobbe et al., 2021) and MATH (Hendrycks et al., 2021) seed datasets. The framework consists of four stages: seed problem filtering, agentic problem generation, reasoning-solution generation, and synthetic-data evaluation. Using AgenticMath, we construct a high-quality math dataset to fine-tune LLMs and enhance their mathematical reasoning ability. All agent prompts are provided in Appendix A.8.

### 3.1   PROBLEM DEFINITION

Given a seed dataset $\mathcal{D}_{\text{seed}} = \{q_i\}_{i=1}^N$, where each $q_i$ denotes an original mathematical problem from MATH (Hendrycks et al., 2021) and GSM8K (Cobbe et al., 2021), we employ large language models (LLMs) to construct a new dataset of problem–solution pairs, eliminating ground-truth labels and reducing costly human annotations. Formally, the transformation can be summarized as $\mathcal{D}_{\text{seed}} \Rightarrow \mathcal{D}_{\text{final}}$, where the resulting dataset is denoted as $\mathcal{D}_{\text{final}} = \{(q_i, r_i)\}_{i=1}^{N'}$. The problem component $q_i$ consists of both original problems from $\mathcal{D}_{\text{seed}}$ and newly synthesized problems, while the solution component $r_i$ is entirely generated by the LLM. This dataset $\mathcal{D}_{\text{final}}$ is subsequently used as training data for supervised fine-tuning (SFT). The SFT objective is to maximize likelihood of the target response given the prompt query. Specifically, the loss function is defined as:

$$\mathcal{L}(\theta) = -\frac{1}{N'} \sum_{i=1}^{N'} \log P\big(r_i \mid q_i; \theta\big), \tag{1}$$

where $\theta$ denotes model parameters, $q_i$ the input problem, and $r_i$ the generated solution.

### 3.2   STAGE 1: SEED PROBLEM FILTERING

Using seed problems as references to synthesize more problems can effectively enhance the model's mathematical capabilities. However, current methods ignore that low-quality, low-difficulty problems in the seed dataset may not be worthy of serving as references. Hence, we propose a training-free and label-free filtering method to identify high-value seed problems.

**LLM-based Scoring.** Each candidate problem from the seed dataset $\mathcal{D}_{\text{seed}} = \{q_i^{\text{seed}}\}_{i=1}^N$ is scored by a large language model (LLM) along three dimensions: *Complexity* ($c_i$), *Information Value* ($v_i$), and *Clarity & Precision* ($p_i$), with ratings in the range $\{0, 1, \ldots, 5\}$. The Evaluator, based on GPT-4o-mini (Achiam et al., 2023), processes a scoring prompt to generate the score list $\mathbf{s}_i = [c_i, v_i, p_i]$. The overall score $\bar{s}_i$ is defined as the arithmetic mean of these three dimensions. As a result of this evaluation, we obtain the scored dataset $\mathcal{D}_{\text{scored}} = \{(q_i^{\text{seed}}, \mathbf{s}_i, \bar{s}_i)\}_{i=1}^N$, where each problem $q_i$ is associated with both its dimension-wise scores $\mathbf{s}_i$ and aggregated score $\bar{s}_i$.

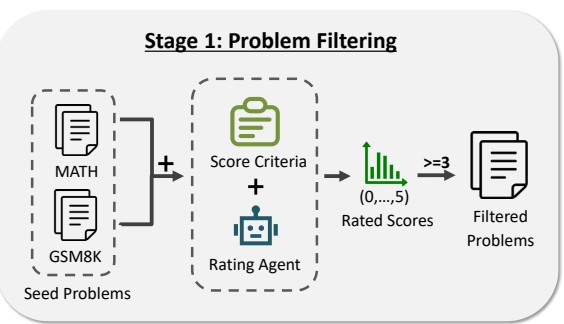

Figure 2: Workflow of Stage 1 showing the filtering process that removes low-quality seed problems and retains high-value ones for subsequent synthesis.

---

**Problem Filtering Dimensions and Prompt Example**

**Complexity:** Does it integrate multiple mathematical domains (e.g., algebra + geometry) or demand critical thinking?

**Information Value:** Does it contain useful knowledge or reasoning opportunities? Can it help learners discover concepts, strategies, or patterns?

**Clarity & Precision:** Is the question unambiguous, logically consistent, and free of errors? Poorly framed questions score lower.

---

**Score Curation.** To mitigate potential rating errors introduced by LLM-based evaluation, we apply a score curation procedure inspired by DS2 (Pang et al., 2024) and the clusterability-based method of (Zhu et al., 2021). Starting from the scored dataset $\mathcal{D}_{\text{scored}}$, we construct a Score Transition Matrix $T$ to capture consistency patterns among neighboring problems in the embedding space. By leveraging $k$-nearest neighbor agreement, problems whose ratings deviate from those of their local neighborhood are adjusted toward more reliable estimates. This process yields the curated dataset $\mathcal{D}_{\text{curated}} = \{(q_i^{\text{seed}}, \tilde{s}_i)\}_{i=1}^N$, where each problem $q_i^{\text{seed}}$ is paired with its corrected overall score $\tilde{s}_i$, representing a refined estimate of problem quality.

**Filtering Rule.** In the final step, we impose a quality threshold of $\tau = 3$ on the curated scores. The resulting dataset $\mathcal{D}_{\text{filter}}$ is derived from $\mathcal{D}_{\text{curated}}$ by retaining only those problems whose corrected overall score $\tilde{s}_i$ meets or exceeds this threshold. This filtering process excludes problems that are overly simplistic, ambiguous, or uninformative, ensuring that the retained problems are well-formed and valuable. The overall pipeline for seed problem filtering can be summarized as: $\mathcal{D}_{\text{seed}} \Rightarrow \mathcal{D}_{\text{scored}} \Rightarrow \mathcal{D}_{\text{curated}} \Rightarrow \mathcal{D}_{\text{filter}}$.

### 3.3 STAGE 2: AGENTIC PROBLEM GENERATION

Although closed-source models can generate complex new problems by following instructions, hallucinations still appear, leading to low-quality or poorly phrased outputs. In multi-agent settings, self-reflection provides a way to correct such errors. Building on this idea, we design a framework for problem synthesis that ensures quality through three roles: a rephrase agent, a review agent, and a revise agent.

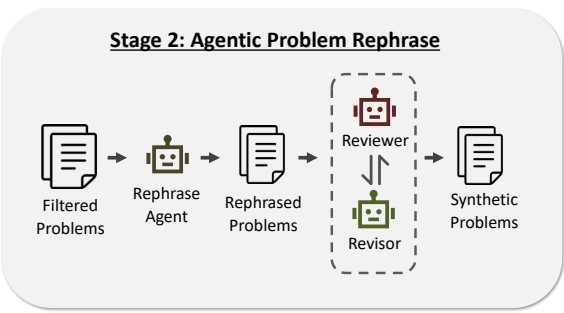

Figure 3: Workflow of Stage 2 showing multi-agent problem generation with Rephrase, Review, and Revise agents, along with iterative refinement to ensure clarity, coherence, and mathematical validity.

**Problem Rephrase Agent.** From the filtered dataset $\mathcal{D}_{\text{filter}} = \{q_i^{\text{seed}}\}_{i=1}^M$, each

problem is expanded into six paraphrased variants by the Problem Rephrase Agent. The new collection is denoted as $\mathcal{D}_{\text{rephrase}} = \{q_i^{\text{rep}}\}_{i=1}^{M'}$, where each $q_i^{\text{rep}}$ corresponds to a rephrased version of its seed problem. Rephrasing is guided by task-specific instructions to GPT-4o-mini, designed to preserve the mathematical intent while introducing greater difficulty, lexical richness, and syntactic diversity.

**Problem Review Agent.** The rephrased dataset $\mathcal{D}_{\text{rephrase}} = \{q_i^{\text{rep}}\}_{i=1}^{M'}$ is passed to the Problem Review Agent for evaluation. Each candidate problem is checked against its original version following a review instruction. The assessment spans three dimensions: *Clarity & Grammar*, *Logical Coherence & Completeness*, and *Mathematical Validity & Solvability*. For every candidate, the agent assigns a score in the range $[1, 5]$ and, if needed, provides textual feedback for improvement. The outcome is the reviewed dataset $\mathcal{D}_{\text{review}} = \{(q_i^{\text{rep}}, \bar{s}_i^{\text{rev}}, a_i^{\text{rev}})\}_{i=1}^{M'}$, where each rephrased problem is paired with its score and an optional suggestion.

> **Problem Review Dimensions and Prompt Example**
>
> **Clarity & Grammar:** The question must be grammatically correct, precisely phrased, and easy to understand. It should avoid ambiguity in wording or phrasing.
> **Logical Coherence & Completeness:** All elements of the problem (e.g., given information, constraints, relationships, objectives) must be logically interconnected and sufficient. The problem should present a clear, sequential path for reasoning, without missing information required for the specified solution approach.
> **Mathematical Validity & Solvability:** The problem must be fundamentally a mathematics problem, with all its premises and conditions being mutually consistent and mathematically sound...

**Problem Revise Agent.** Based on the reviewed dataset $\mathcal{D}_{\text{review}}$, the Problem Revise Agent targets rephrased problems with scores below the threshold $\tau_{\text{rev}} = 4.5$. For each low-scoring case, the problem $q_i^{\text{rep}}$ is revised according to reviewer feedback $a_i^{\text{rev}}$. This step corrects issues such as unclear phrasing, weak logical flow, or invalid mathematical conditions. The result is the revised dataset $\mathcal{D}_{\text{revise}} = \{q_i^{\text{rev}}\}_{i=1}^{M''}$, which retains only problems that reach the required quality level.

**Problem Review–Revise Interaction.** To further strengthen problem quality, an iterative loop between the Review and Revise agents is applied. Starting from $\mathcal{D}_{\text{review}}$, all problems scoring below $\tau_{\text{rev}}$ enter this refinement process. In each round, the Review agent re-evaluates a candidate, assigns a new score, and may suggest specific improvements. The Revise Agent incorporates this feedback to produce an updated version. The loop runs for at most three iterations, with early stopping once the threshold is met. Afterward, only problems with final scores above 4.5 are kept, while the rest are discarded. The outcome is the refined dataset $\mathcal{D}_{\text{refined}} = \{q_i^{\text{ref}}\}_{i=1}^{K}$, containing high-quality rephrasings that meet the required standard.

### 3.4 STAGE 3: SOLUTION GENERATION

**Solution Solver Agent.** To provide high-quality reasoning traces for training, we employ a one-shot Chain-of-Thought (CoT) prompting scheme that elicits multi-step reasoning solution paths. The Solver Agent works on two distinct datasets: the original seed problems $\mathcal{D}_{\text{seed}} = \{q_i^{\text{seed}}\}_{i=1}^{N}$ and the refined rephrased problems $\mathcal{D}_{\text{ref}} = \{q_j^{\text{ref}}\}_{j=1}^{K}$. For each problem, GPT-4o-mini is prompted with a single CoT exemplar to generate a detailed, step-by-step solution. This process produces two corresponding solution-augmented datasets:

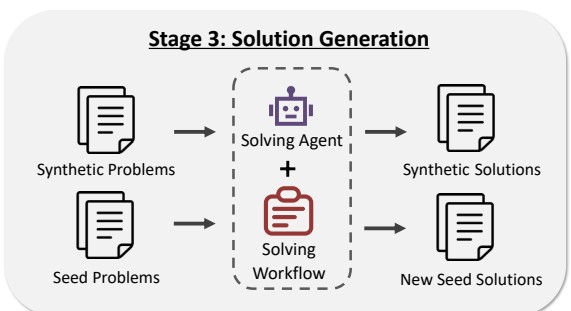

Figure 4: Workflow of Stage 3 showing solution generation by the Solver Agent with step-by-step reasoning.

$$\mathcal{D}_{\text{seed}}^{\text{sol}} = \{(q_i^{\text{seed}}, a_i^{\text{sol}})\}_{i=1}^{N}, \quad \mathcal{D}_{\text{ref}}^{\text{sol}} = \{(q_j^{\text{ref}}, a_j^{\text{sol}})\}_{j=1}^{K},$$

where every problem from $\mathcal{D}^{\text{seed}}$ and $\mathcal{D}^{\text{ref}}$ is paired with a synthetic solution $a^{\text{sol}}$ that explicitly demonstrates the intermediate reasoning steps.

### 3.5 STAGE 4: SYNTHETIC DATA EVALUATION

In this stage, the scoring and curation framework from Stage 1 is extended to problem–solution pairs. The evaluation targets the synthetic set $\mathcal{D}^{\text{sol}}_{\text{ref}} = \{(q^{\text{ref}}_j, a^{\text{sol}}_j)\}_{j=1}^{K}$. Each pair is judged along three dimensions—clarity of the problem, correctness of the solution, and completeness of reasoning. Scores are further stabilized using the Score Transition Matrix and refined through $k$-NN consistency checks. To build a high-quality and diverse subset, a ranking-based selection is applied instead of a fixed threshold. Pairs are first sorted by quality and grouped into

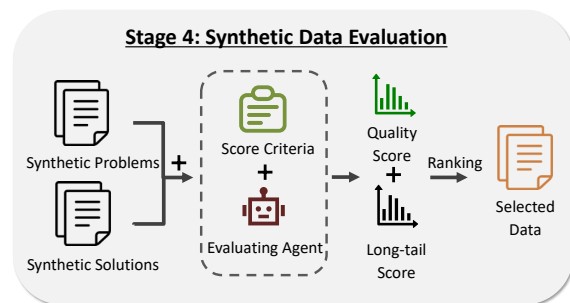

Figure 5: Workflow of Stage 4 showing evaluation of problem–solution pairs for quality and diversity.

discrete score levels. We first group all samples by quality score (from 5 down to 0) and select groups in descending order. When the remaining quota falls within a group larger than needed, we rank samples inside that group using the long-tail diversity score. This strategy ensures that we always take the highest-quality data available while promoting diversity when selecting from oversized groups. This yields the curated dataset $\mathcal{D}^{\text{sol}}_{\text{selected}} = \{(q^{\text{ref}}_j, a^{\text{sol}}_j)\}_{j=1}^{L}$. The final training dataset combines this curated set with the seed-based solutions: $\mathcal{D}_{\text{final}} = \mathcal{D}^{\text{sol}}_{\text{selected}} \cup \mathcal{D}^{\text{sol}}_{\text{seed}}$, ensuring both rigor and diversity for downstream fine-tuning.

## 4 EXPERIMENTS

### 4.1 EXPERIMENTAL SETUP

**Data Synthesis:** We employed GPT-4o-mini (2024-07-18) (Achiam et al., 2023), following (Pei et al., 2025), for all agents across the four stages, including evaluation scoring, problem synthesis, and solution synthesis. Seed problems were sourced from the MATH (Hendrycks et al., 2021) and GSM8K (Cobbe et al., 2021) datasets. For the 30K setting, the final dataset consists of 15K seed problems and 15K AgenticMath-synthesized problems. In Stage 1, we filtered seed problems with scores above 3. In Stage 2, each seed problem was expanded into six rephrased variants, with a review–revise loop requiring scores above 4.5 and running up to three iterations, keeping only those exceeding the threshold. In Stage 4, we applied ranking-based selection with a target of 15K high-quality problem–solution pairs. During all data generation steps, we used a temperature of 0.7 and a maximum token length of 4096.

**Training:** We perform standard instruction tuning on the proposed AgenticMathQA. Following DART-Math (Tong et al., 2024) and Mathfusion (Pei et al., 2025), experiments cover both math-specialized and general base models. For the math-specialized category, we use DeepSeekMath-7B (sha, 2024); for general models, we fine-tune Qwen2.5-3B (Team, 2024), Mistral-7B (Jiang et al., 2023), and Llama3-8B (Grattafiori et al., 2024). The 30K dataset is built from 15K seed problems (sourced from GSM8K and MATH) with corresponding solutions, together with 15K AgenticMath-synthesized problem–solution pairs. Scaling to larger sizes is achieved by augmenting each 30K problem with multiple solutions: duplicating once yields 60K (30K×2), and duplicating twice yields 90K (30K×3). More training details are provided in Appendix A.2.

**Evaluation:** Following DART-Math (Tong et al., 2024) and MathFusion (Pei et al., 2025), we evaluate on six benchmarks spanning both in-domain and out-of-domain (OOD) settings. The in-domain benchmarks are GSM8K (Cobbe et al., 2021) and MATH (Hendrycks et al., 2021), while the OOD benchmarks include CollegeMath (Tang et al., 2024), DeepMind-Mathematics (Saxton et al., 2019), OlympiadBench-Math (He et al., 2024), and TheoremQA (Chen et al., 2023). Further benchmark details are provided in the Appendix A.3.

| Model | # Samples | In-Domain | | Out-of-Domain | | | | AVG |
|---|---|---|---|---|---|---|---|---|
| | | MATH | GSM8K | College | DM | Olympiad | Theorem | |
| Qwen2.5-3B (3–8B General Base Model) | | | | | | | | |
| Qwen2.5-3B-RefAug [†] | 30K | 40.9 | 69.7 | 32.4 | 42.5 | 10.7 | 11.4 | 34.6 |
| Qwen2.5-3B-MathFusion *(Sequential)*[†] | 30K | 39.9 | 72.1 | 28.9 | 50.0 | 23.0 | 14.6 | 38.1 |
| AgenticMath-Qwen2.5-3B | 30K | **62.0** | **83.4** | **46.8** | **72.8** | **25.6** | **31.4** | **53.7** |
| Qwen2.5-3B-MetaMath[†] | 60K | 43.4 | 79.8 | 34.5 | 46.3 | 11.3 | 19.0 | 39.1 |
| Qwen2.5-3B-MMIQC[†] | 60K | 47.3 | 78.2 | 35.6 | 51.2 | 14.7 | 17.1 | 40.7 |
| Qwen2.5-3B-DART-Math[†] | 60K | 53.9 | **84.3** | 42.3 | 59.2 | 18.4 | 26.4 | 47.4 |
| Qwen2.5-3B-MathFusion[†] | 60K | 40.5 | 72.7 | 29.1 | 52.4 | 25.5 | 15.3 | 39.3 |
| AgenticMath-Qwen2.5-3B | 60K | **62.4** | 83.6 | **46.3** | **74.3** | **27.3** | **29.3** | **53.9** |
| DeepSeekMath (7B Math-Specialized Base Model) | | | | | | | | |
| DeepSeekMath-7B-RefAug | 30K | 32.1 | 71.2 | 26.0 | 38.4 | 10.1 | 14.4 | 32.0 |
| DeepSeekMath-7B-MathFusion *(Sequential)* | 30K | 49.9 | 76.6 | 38.8 | 64.6 | **21.6** | 22.8 | 45.7 |
| AgenticMath-DSMath-7B | 30K | **52.4** | **80.1** | **42.6** | **66.8** | 18.2 | **26.9** | **47.8** |
| DeepSeekMath-7B-MetaMath | 60K | 40.0 | 79.0 | 33.2 | 45.9 | 9.5 | 18.9 | 37.8 |
| DeepSeekMath-7B-MMIQC | 60K | 26.3 | 60.6 | 19.2 | 41.5 | 10.4 | 6.8 | 27.5 |
| DeepSeekMath-7B-RefAug | 60K | 33.1 | 71.6 | 26.2 | 35.4 | 10.5 | 14.0 | 31.8 |
| DeepSeekMath-7B-DART-Math | 60K | 51.4 | **82.9** | 39.1 | 62.8 | 21.0 | **27.4** | 47.4 |
| DeepSeekMath-7B-MathFusion | 60K | 53.4 | 77.9 | 39.8 | 65.8 | **23.3** | 24.6 | 47.5 |
| AgenticMath-DSMath-7B | 60K | **55.0** | 80.1 | **43.6** | **69.9** | 20.0 | 27.0 | **49.3** |
| Mistral-7B (3–8B General Base Model) | | | | | | | | |
| Mistral-7B-RefAug | 30K | 15.1 | 61.1 | 10.4 | 15.4 | 3.1 | 11.0 | 19.4 |
| Mistral-7B-MathFusion *(Sequential)* | 30K | 32.7 | 73.9 | 18.9 | 29.3 | 9.3 | 15.5 | 29.9 |
| AgenticMath-Mistral-7B | 30K | **35.3** | **79.5** | **27.0** | **41.9** | **11.9** | **19.3** | **35.8** |
| Mistral-7B-MetaMath | 60K | 22.7 | 70.8 | 14.1 | 27.2 | 5.0 | 12.2 | 25.3 |
| Mistral-7B-MMIQC | 60K | 17.3 | 61.4 | 11.1 | 13.5 | 5.0 | 5.9 | 19.0 |
| Mistral-7B-RefAug | 60K | 17.4 | 63.1 | 12.5 | 18.1 | 3.9 | 11.1 | 21.0 |
| Mistral-7B-DART-Math | 60K | 34.1 | 77.2 | 23.4 | 36.0 | 8.7 | 18.2 | 32.9 |
| Mistral-7B-MathFusion | 60K | **41.6** | 79.8 | 24.3 | 39.2 | **13.6** | 18.1 | 36.1 |
| AgenticMath-Mistral-7B | 60K | 39.5 | **82.3** | **28.7** | **47.1** | 12.4 | **20.5** | **38.4** |
| Llama3-8B (3–8B General Base Model) | | | | | | | | |
| Llama3-8B-RefAug | 30K | 20.8 | 67.3 | 15.7 | 25.9 | 4.7 | 13.6 | 24.7 |
| Llama3-8B-MathFusion *(Sequential)* | 30K | 38.8 | 77.9 | 25.1 | **42.0** | **12.6** | 17.0 | 35.6 |
| AgenticMath-Llama3-8B | 30K | **36.8** | **78.4** | **29.6** | 40.3 | 11.4 | **20.4** | **36.2** |
| Llama3-8B-MetaMath | 60K | 28.7 | 78.5 | 19.7 | 31.3 | 5.3 | 16.1 | 29.9 |
| Llama3-8B-MMIQC | 60K | 24.4 | 69.7 | 13.4 | 30.9 | 5.2 | 10.6 | 25.7 |
| Llama3-8B-RefAug | 60K | 20.3 | 68.6 | 15.5 | 29.1 | 5.5 | 13.0 | 25.3 |
| Llama3-8B-DART-Math | 60K | 39.6 | **82.2** | 27.9 | 36.9 | 12.9 | **22.9** | 37.6 |
| Llama3-8B-MathFusion | 60K | **46.5** | 79.2 | 27.9 | 43.4 | **17.2** | 20.0 | 39.0 |
| AgenticMath-Llama3-8B | 60K | 40.4 | 80.1 | **31.6** | **46.7** | 14.1 | 22.6 | **39.3** |

Table 1: Evaluation results across in-domain and out-of-domain math benchmarks with 30K–60K samples. Most baseline results are reported from (Pei et al., 2025), while entries marked with † denote results reproduced by us. **Bold** numbers indicate the best performance within the same type of sample size and base model. Rows highlighted in blue correspond to our AgenticMath results.

**Baseline:** We compare AgenticMath with state-of-the-art methods in two settings. For large-scale training, we include MetaMath (Yu et al., 2023), RFT (Yuan et al., 2023), DART-Math (Tong et al., 2024), MathScale (Tang et al., 2024), DeepSeekMath-7B-Instruct (sha, 2024), RefAug (Zhang et al., 2024b), MMIQC (Liu et al., 2025a), and WizardMath (Luo et al., 2023), all using 400K–2.3M samples. For small-scale, we evaluate 30K and 60K subsets: RefAug and MathFusion provide native 30K versions, while other baselines are randomly down-sampled to 60K from the large-scale dataset above.

## 4.2 MAIN RESULTS

**AgenticMath Achieves SOTA Performance at 30K–60K Data Scale.** Table 1 shows that AgenticMath consistently outperforms all baselines at both 30K and 60K scales. Across every base model (Qwen2.5-3B, DeepSeekMath-7B, Mistral-7B, and Llama3-8B), our method achieves the highest average score and sets new state-of-the-art performance under small-scale training. For example, with 30K samples, AgenticMath-Qwen2.5-3B reaches 53.7 average accuracy, surpassing MathFusion by over 15 points. At 60K, AgenticMath continues to improve and outperforms all other baseline methods trained with the same number of samples. These results demonstrate that rigorous multi-agent synthesis and quality control provide significantly better data efficiency than prior methods.

| Model | # Samples | In-Domain | | Out-of-Domain | | | | AVG |
|---|---|---|---|---|---|---|---|---|
| | | MATH | GSM8K | College | DM | Olympiad | Theorem | |
| **DeepSeekMath (7B Math-Specialized Base Model)** | | | | | | | | |
| DeepSeekMath-7B-RFT | 590K | 53.0 | **88.2** | 41.9 | 60.2 | 19.1 | 27.2 | 48.3 |
| DeepSeekMath-7B-DART-Math | 590K | 53.6 | 86.8 | 40.7 | 61.6 | 21.7 | **32.2** | 49.4 |
| DeepSeekMath-7B-Instruct | 780K | 46.9 | 82.7 | 37.1 | 52.2 | 14.2 | 28.1 | 43.5 |
| DeepSeekMath-7B-MMIQC | 2.3M | 45.3 | 79.0 | 35.3 | 52.9 | 13.0 | 23.4 | 41.5 |
| DeepSeekMath-7B-MathFusion-7B | 195K | **58.2** | 79.5 | 40.3 | 69.1 | **25.5** | 27.0 | **49.9** |
| AgenticMath-DSMath-7B | 30K | 52.4 | 80.1 | 42.6 | 66.8 | 18.2 | 26.9 | 47.8 |
| AgenticMath-DSMath-7B | 60K | 55.0 | 80.1 | **43.6** | **69.9** | 20.0 | 27.0 | 49.3 |
| **Mistral-7B (3–8B General Base Model)** | | | | | | | | |
| Mistral-7B-MetaMath | 400K | 29.8 | 76.5 | 19.3 | 28.0 | 5.9 | 14.0 | 28.9 |
| Mistral-7B-WizardMath-V1.1 | 418K | 32.3 | 80.4 | 23.1 | 38.4 | 7.7 | 16.6 | 33.1 |
| Mistral-7B-RFT | 590K | 38.7 | **82.3** | 24.2 | 35.6 | 8.7 | 16.2 | 34.3 |
| Mistral-7B-DART-Math | 590K | **45.5** | 81.1 | **29.4** | 45.1 | **14.7** | 17.0 | **38.8** |
| Mistral-7B-MathScale | 2.0M | 35.2 | 74.8 | 21.8 | – | – | – | – |
| Mistral-7B-MMIQC | 2.3M | 37.4 | 75.4 | 28.5 | 38.0 | 9.4 | 16.2 | 34.2 |
| AgenticMath-Mistral-7B | 30K | 35.3 | 79.5 | 27.0 | 41.9 | 11.9 | 19.3 | 35.8 |
| AgenticMath-Mistral-7B | 60K | 39.5 | **82.3** | 28.7 | **47.1** | 12.4 | **20.5** | 38.4 |
| **Llama3-8B (3–8B General Base Model)** | | | | | | | | |
| Llama3-8B-MetaMath | 400K | 32.5 | 77.3 | 20.6 | 35.0 | 5.5 | 13.8 | 30.8 |
| Llama3-8B-RFT | 590K | 39.7 | **81.7** | 23.9 | 41.7 | 9.3 | 14.9 | 35.2 |
| Llama3-8B-MMIQC | 2.3M | 39.5 | 77.6 | 29.5 | 41.0 | 9.6 | 16.2 | 35.6 |
| Llama3-8B-DART-Math | 590K | **46.6** | 81.1 | 28.8 | 48.0 | **14.5** | 19.4 | 39.7 |
| AgenticMath-Llama3-8B | 30K | 36.8 | 78.4 | 29.6 | 40.3 | 11.4 | 20.4 | 36.2 |
| AgenticMath-Llama3-8B | 60K | 40.4 | 80.1 | 31.6 | 46.7 | 14.1 | **22.6** | 39.3 |
| AgenticMath-Llama3-8B | 90K | 42.8 | 81.4 | **33.0** | **48.6** | 13.9 | 21.8 | **40.3** |

Table 2: Results on math benchmarks comparing AgenticMath (30K/60K/90K) with large-scale baselines trained on 400K–2.3M data. All baseline results are reported from their respective papers. **Bold** numbers indicate the best performance within the same type of base model. Rows highlighted in blue correspond to our AgenticMath results.

**AgenticMath Matches or Surpasses Larger-Scale Baselines with Much Less Data.** Table 2 shows that AgenticMath, even with only 30K–90K samples, matches or surpasses baselines trained on hundreds of thousands or even millions of samples. For example, AgenticMath-DSMath-7B (60K) achieves an average score of 49.3, close to DeepSeekMath-7B-MathFusion (195K, 49.9) and higher than DeepSeekMath-7B-RFT (590K, 48.3). On general models, AgenticMath-Mistral-7B (60K) reaches 38.4, comparable to Mistral-7B-DART-Math (590K, 38.8) and outperforming Mistral-7B-RFT (590K, 34.3). Most notably, AgenticMath-Llama3-8B (90K) achieves 40.3, surpassing Llama3-8B-DART-Math (590K, 39.7) and all other large-scale Llama3 baselines. These results confirm that AgenticMath delivers competitive or superior performance with much fewer samples, highlighting its strong data efficiency.

### 4.3 Understanding AgenticMath: Ablations and Insights

**All Modules Contribute to Performance Gains.** To more clearly quantify the contribution of each module in the Agentic-Math pipeline, we perform ablation studies on Mistral-7B using a fixed training set of 15K synthesized samples. As shown in Table 3, the pipeline exhibits consistent, stage-by-stage improvements. The initial Problem Rephrase step provides a strong baseline (31.4 AVG). Introducing Seed Filtering further improves performance by removing low-quality or noisy seed questions, yielding a +0.6 gain. The subsequent Problem Review–Revise loop brings the largest additional improvement (+1.0), demonstrating the importance of iterative refinement for producing clearer and more logically coherent problems. Finally, the Synthetic Data Evaluation stage contributes a modest but steady boost (+0.2), as most low-quality samples have already been filtered out in earlier stages. Overall, these results show that the full pipeline is cumulatively beneficial, and that each stage plays a meaningful role in improving data quality and downstream model performance.

Table 3: Ablation study on the contribution of different pipeline stages.

| Method | Samples | AVG |
|---|---|---|
| Problem Rephrase | 15K | 31.4 |
| + Seed Filtering | 15K | 32.0 ↑ 0.6 |
| + Problem Review–Revise | 15K | 33.0 ↑ 1.0 |
| + Synthetic Data Evaluation | 15K | **33.2** ↑ 0.2 |

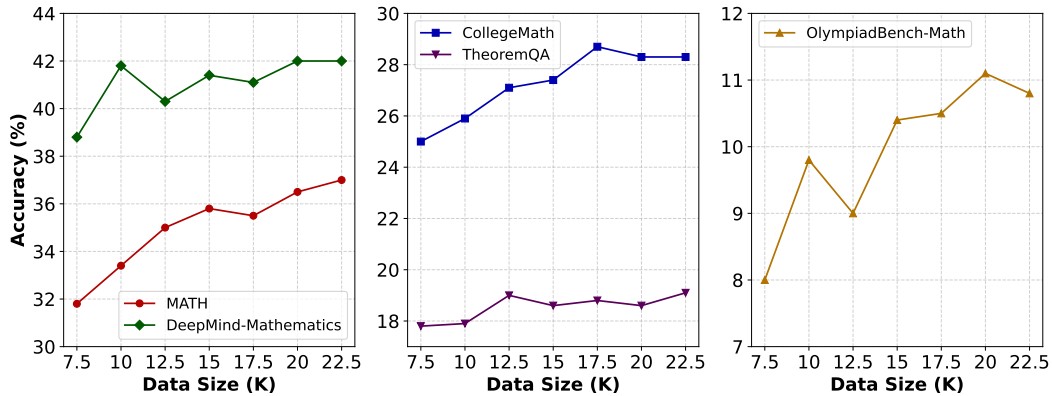

Figure 6: Llama3-8B performance across benchmarks as training size increases.

**Problem Quality Directly Boosts Performance.** We further investigate the impact of different filtering thresholds in Stage 1 using Mistral-7B as the base model. Table 4 shows that the use of higher thresholds for the filtering of seed problems leads to better results, confirming that the quality of the selected problems directly impacts the performance of reasoning. Although our main experiments adopt a threshold of score 3, increasing it to score 4 yields further gains. This indicates that AgenticMath benefits from stricter quality control and still offers further optimization space for even stronger performance.

Table 4: Performance with different thresholds for seed problem filtering.

| Threshold | Samples | AVG |
|---|---|---|
| Score = 2 | 30K | 33.4 |
| Score = 3 | 30K | 34.9 |
| Score = 4 | 30K | **35.0** |

**Larger Training Samples Yield Stronger Reasoning Performance.** We analyze how varying the amount of training data affects model performance. Starting from a base setting with 7.5K MATH samples, we gradually add synthetic data in increments of 2.5K, up to a total of 22.5K samples. As shown in Figure 6, Llama3-8B shows consistent accuracy gains on different benchmarks as the dataset grows, confirming a strong positive correlation between training size and reasoning ability. This upward trend demonstrates that increasing data with our multi-agent framework steadily strengthens performance.

**Illustrative Cases of Enhanced Problem Quality** Appendix A.7 provides several illustrative cases refined by the Reviewer and Revise Agents, showing how our method improves clarity and correctness of mathematical problems.

## 5 CONCLUSION

In this work, we introduced *AgenticMath*, a multi-agent framework for high-quality synthetic data generation of mathematical problems and solutions. By coordinating agents for filtering, rephrasing, revision, solution generation, and joint evaluation, *AgenticMath* provides a systematic and scalable approach to generating high-quality math reasoning data. The resulting dataset, *AgenticMathQA*, is released in curated 30K, 60K, and 90K versions, emphasizing clarity, correctness, and diversity rather than data scale. Extensive experiments across multiple open-source base models show that with only 5%–15% of the data size scale, *AgenticMath* matches or surpasses methods trained on 400K–2.3M samples, achieving SOTA performances by referring to baselines with the same data scale. These results highlight that data quality—supported by rigorous multi-agent design—plays a more decisive role than dataset size in advancing mathematical reasoning in large language models.

## ETHICS STATEMENT

This work does not involve human subjects or sensitive personal data. All datasets used are publicly available (MATH and GSM8K), and our synthetic data generation follows the ICLR Code of Ethics by avoiding the release of harmful or biased content. The proposed framework, AgenticMath, is intended purely for advancing research in mathematical reasoning and does not introduce applications that could cause societal harm. We release our curated datasets and code in compliance with licensing terms to ensure transparency, reproducibility, and fair access for the research community.

## REPRODUCIBILITY STATEMENT

We have taken multiple steps to ensure reproducibility. Details of the multi-agent pipeline design, training and evaluation are provided in the main text. Additional training details, hyperparameters are included in the Appendix. All benchmarks used (MATH, GSM8K, CollegeMath, DeepMind-Mathematics, OlympiadBench, and TheoremQA) are publicly available. The full AgenticMathQA dataset and implementation code will be released upon publication to enable independent verification of our results.

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

# A APPENDIX

## A.1 THE USE OF LARGE LANGUAGE MODELS (LLMS)

In this work, LLMs are used to assist with text revision and grammar refinement, ensuring concise and fluent writing. LLMs further support formatting adjustments for figures and tables, improving readability and consistency across the paper. LLMs are also applied to refine mathematical notation, adjust formula symbols, and standardize technical expressions, helping maintain clarity and precision throughout the manuscript. LLMs serve only as auxiliary tools, with all final decisions and edits made by the authors.

## A.2 TRAINING AND INFERENCE DETAILS

All models—including our baseline reproductions—are fine-tuned for 3 epochs using a global batch size of 96 on 6×NVIDIA A800 GPUs. We adopt a peak learning rate of 1e-6 (5e-6 for DeepSeekMath-7B), combined with a linear warm-up over the first 3% of steps and cosine decay thereafter. The maximum sequence length is fixed at 4096 tokens.

During inference, we fix the sampling temperature to 0 to ensure deterministic outputs, and set the maximum generation length (max tokens) to 2048 for all models. We use a fixed random seed of 0 for reproducibility and set the number of inference trials to 1 for every evaluation. For our primary models, we adopt a standard Chain-of-Thought (CoT) prompting scheme. Specifically:

- **Training prompt:** `Question:  {problem} Answer:`
- **Evaluation prompt:** `Question:  {problem} Answer:  Let's think step by step.`

This prompt design follows common practice in mathematical reasoning tasks and encourages the model to generate explicit intermediate reasoning steps. For Mistral 7B and Llama 3 8B, we instead use the Alpaca instruction-following template during inference:

```
Below is an instruction that describes a task.  Write
a response that appropriately completes the request:
### Instruction:
{problem}
### Response:
```

We adopt this template because our preliminary experiments showed that Alpaca-style instructions consistently yield better reasoning quality on these two architectures compared with the CoT-style prompt. This observation is also aligned with the findings reported in MathFusion, where Alpaca-style prompting was similarly found to be more effective.

## A.3 EVALUATION BENCHMARKS

We provide detailed descriptions of the six benchmarks used in our evaluation:

**In-Domain:** (i) GSM8K (Cobbe et al., 2021), consisting of grade-school arithmetic word problems that are relatively simple. (ii) MATH (Hendrycks et al., 2021), a large-scale dataset of competition-level problems that are significantly more challenging.

**Out-of-Domain (OOD):** (i) CollegeMath (Tang et al., 2024), with 2,818 college-level problems drawn from nine textbooks across seven domains (e.g., linear algebra, differential equations), designed to test generalization to complex mathematics. (ii) DeepMind-Mathematics (Saxton et al., 2019), a collection of 1,000 problems covering a national school curriculum (up to age 16), assessing basic reasoning across varied types. (iii) OlympiadBench-Math (He et al., 2024), providing 675 Olympiad-level problems (English text-only subset) targeting the most challenging reasoning tasks. (iv) TheoremQA (Chen et al., 2023), consisting of 800 problems that require applying mathematical theorems across mathematics, physics, and engineering, testing theoretical reasoning in STEM.

Table 5: 10K-sample comparison using the same teacher model.

| Dataset | # Samples | MATH | GSM8K | College | DM | Olympiad | Theorem | AVG |
|---|---|---|---|---|---|---|---|---|
| MetaMath | 10K | 24.9 | 70.4 | 19.0 | 28.5 | 5.1 | 13.8 | 26.9 |
| DARTMath | 10K | 29.3 | 66.4 | 21.3 | 37.4 | 8.7 | 16.7 | 29.9 |
| ScaleQuest | 10K | 24.9 | 66.4 | 17.5 | 27.3 | 7.7 | 14.1 | 26.3 |
| AgenticMath (ours) | 10K | **29.6** | **73.8** | **24.6** | **38.2** | 7.4 | 16.3 | **31.6** |

### A.4 FAIRNESS CONCERNS REGARDING TEACHER MODELS

For further strengthen the fairness discussion, we conduct an additional 10K-sample controlled comparison across MetaMath, DARTMath, ScaleQuest, and our AgenticMath, all trained under the same SFT configuration (Mistral-7B) and, where applicable, using the same teacher model (GPT-4o-mini, 2024-07-18) for solution generation. As shown in Table 5, AgenticMath achieves the highest average performance among all methods. This controlled experiment confirms that the observed improvements do not arise from using a stronger teacher model—since all datasets share the same teacher—but rather from the design of our synthesis pipeline itself.

#### A.4.1 SENSITIVITY ANALYSIS ON THE REVISE THRESHOLD

To further examine the impact of the revise threshold, we conduct a sensitivity study with $\tau_{rev} \in \{3.5, 4.0, 4.5\}$ using Llama3-8B under a fixed 30K SFT setting. The results are reported in Table 6.

Table 6: Sensitivity analysis of the revise threshold $\tau_{rev}$ using Llama3-8B.

| $\tau_{rev}$ | Samples | MATH | GSM8K | College | DM | Olympiad | Theorem | AVG |
|---|---|---|---|---|---|---|---|---|
| 4.5 | 30K | 36.8 | 78.4 | 29.6 | 40.3 | 11.4 | 20.4 | **36.2** |
| 4.0 | 30K | 36.6 | 77.5 | 28.2 | 43.1 | 11.5 | 20.0 | **36.2** |
| 3.5 | 30K | 37.8 | 77.4 | 27.4 | 41.0 | 10.3 | 20.0 | 35.7 |

As shown in Table 6, the settings $\tau_{rev} = 4.0$ and $4.5$ produce highly consistent results, demonstrating that the review–revise mechanism remains stable across reasonable threshold choices. In contrast, $\tau_{rev} = 3.5$ yields lower overall performance, which is expected since a looser threshold admits more low-quality candidates into subsequent stages, ultimately weakening the final dataset quality.

#### A.4.2 ANALYSIS OF THE THREE REVIEW–REVISE ITERATIONS.

To further address the reviewer's question regarding the choice of three review–revise iterations, we evaluate the quality of the problems that pass each round using GPT-4o-mini. Specifically, we compute the average *complexity*, *information value*, and *clarity* scores for all accepted problems after each iteration.

Table 7: Quality metrics across three review–revise iterations.

| Metric | Round 1 | Round 2 | Round 3 |
|---|---|---|---|
| Complexity | 3.86 | 3.93 | 3.92 |
| Information Value | 3.96 | 4.03 | 4.02 |
| Clarity | 4.35 | 4.44 | 4.45 |
| **Avg** | **4.06** | **4.13** | **4.13** |

As shown in Table 7, the first two review–revise iterations produce consistent improvements across all metrics. By the third iteration, however, the gains largely stabilize, indicating diminishing returns. This analysis supports our design choice of using three iterations: it captures most of the quality improvements while avoiding unnecessary computation beyond the point of saturation.

### A.5 ADDITIONAL ANALYSIS OF SYNTHETIC DATA QUALITY AND CHARACTERISTICS

To provide further quantitative evidence of the quality and semantic composition of the synthesized data, we conduct a post-hoc analysis using GPT-4o-mini as an external evaluator. Specifically, we (i) assign quality scores to the refined problems and (ii) classify each problem into standard mathematical topics.

#### A.5.1 QUALITY SCORE DISTRIBUTION

Table 8 reports the quality distribution assigned by GPT-4o-mini over the 18,679 refined problems. The majority of synthesized questions receive a score of $\geq 4$, indicating strong clarity, coherent reasoning, and non-trivial complexity across the dataset.

Table 8: Quality score distribution of 18,679 refined problems, evaluated by GPT-4o-mini.

| Quality Score | Count | Percentage |
|:---:|:---:|:---:|
| 1 | 307 | 1.64% |
| 2 | 1175 | 6.29% |
| 3 | 5053 | 27.05% |
| 4 | 12142 | 65.00% |
| 5 | 2 | 0.01% |

#### A.5.2 TOPIC DISTRIBUTION OF THE FINAL 15K DATASET

GPT-4o-mini is further used to classify each problem in the final 15,000-sample dataset into standard mathematical domains. The resulting topic distribution is shown in Table 9. The distribution demonstrates broad semantic coverage across major mathematical disciplines, with strong representation in combinatorics, geometry, algebra, and number theory.

Table 9: Topic distribution of the final 15K synthetic dataset (classified by GPT-4o-mini).

| Topic | Count | Percentage |
|:---|:---:|:---:|
| Counting & Probability | 3705 | 24.70% |
| Geometry | 3326 | 22.17% |
| Algebra | 2446 | 16.31% |
| Number Theory | 1475 | 9.83% |
| Calculus | 1470 | 9.80% |
| Precalculus | 1456 | 9.71% |
| Intermediate Algebra | 907 | 6.05% |
| Prealgebra | 123 | 0.82% |
| Linear Algebra | 71 | 0.47% |
| Others | 21 | 0.14% |

### A.6 DETAILED STATISTICS OF THE DATA GENERATION PIPELINE

To provide a clearer understanding of the robustness and transparency of our data generation pipeline, we report detailed statistics for all major stages, including seed scoring, rephrase expansion, the multi-round review–revise refinement process, and the final data evaluation. These analyses illustrate how the pipeline progressively improves *problem diversity, clarity, and complexity*, which are key for enhancing downstream mathematical reasoning ability.

### A.6.1 SEED DATA SCORING

We first evaluate all raw seed questions using our label-free scoring mechanism. A total of 7,001 questions satisfy the filtering threshold (score $\geq$ 3). The full distribution is shown in Table 10. This stage ensures that only seed questions with sufficient structural soundness and baseline complexity are used for further synthesis.

Table 10: Score distribution for GSM8K and MATH seed datasets.

| Dataset | Score=0 | Score=1 | Score=2 | Score=3 | Score=4 | Score=5 | Score $\geq$ 3 |
|---------|---------|---------|---------|---------|---------|---------|----------------|
| GSM8K   | 823     | 3522    | 1849    | 1188    | 91      | 0       | 1279           |
| MATH    | 69      | 825     | 884     | 2053    | 3415    | 254     | 5722           |

### A.6.2 REPHRASE EXPANSION

To enhance problem *complexity* and *diversity* while preserving core semantics, each filtered seed question is rephrased six times. All 42,006 synthesized candidates proceed to the review–revise process. This expands the problem pool as follows:

Table 11: Rephrase expansion of filtered seed questions.

| Stage | Count Calculation | Total |
|-------|-------------------|-------|
| Rephrase Expansion | $(1279 + 5722) \times 6$ | 42,006 |

### A.6.3 REVIEW–REVISE LOOP

Our three-round review–revise process progressively improves *clarity*, *logical correctness*, and *mathematical validity*. Across rounds, vague or low-quality questions are removed, while clearer and more coherent problems are retained. Table 12 summarizes the filtering behavior across rounds.

Table 12: Statistics of the three-round review–revise refinement.

| Round | Total Inputs | Passed | Pass Rate |
|-------|--------------|--------|-----------|
| 1     | 42,006       | 7,438  | 17.71%    |
| 2     | 34,568       | 6,526  | 18.88%    |
| 3     | 28,042       | 4,718  | 16.83%    |
| **All Rounds** | –   | **18,682** | –    |

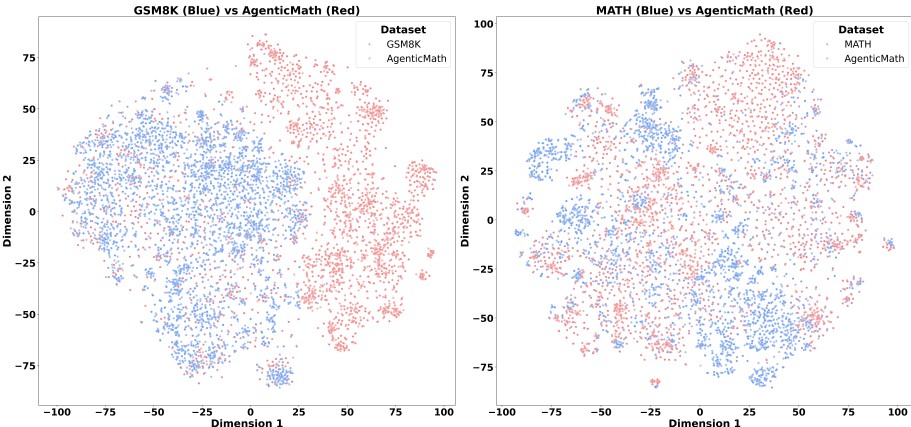

Figure 7: t-SNE Semantic Distribution.

### A.6.4 SYNTHETIC DATA QUALITY DISTRIBUTION

After refinement, 18,679 high-quality synthetic problems remain. The fact that 65% of the questions are assigned a score of 4, with another 27% scoring 3, demonstrates that the majority of synthesized problems exhibit strong clarity, coherent reasoning, and meaningful complexity. Their quality distribution (evaluated by GPT-4o-mini) is shown in Table 13.

Table 13: Quality score distribution of 18,679 refined synthetic problems.

| Score | Count | Percentage |
|-------|-------|-----------|
| 1 | 307 | 1.64% |
| 2 | 1175 | 6.29% |
| 3 | 5053 | 27.05% |
| 4 | 12,142 | 65.00% |
| 5 | 2 | 0.01% |

### A.6.5 FINAL DATASET CONSTRUCTION

To construct the final 15K dataset used in our experiments, we jointly rank all refined samples using a combination of the quality score and the long-tail diversity score. This ranking procedure prioritizes both *overall quality* and *distributional diversity*. The top 15,000 problems from this ranked list form the final synthetic dataset.

## A.7 EXAMPLES OF PROBLEM REVISION

### A.7.1 EXAMPLES OF PROBLEM REVISIONS FROM GSM8K

---

**Example of problem refinement by the Reviewer and Revise Agents.**

**Original Seed Problem:** Ivory got four more riddles than Josh did. Taso got twice as many riddles as Ivory did. If Josh has 8 riddles, how many riddles does Taso have?

**Rephrased Problem:** Josh is participating in a school riddle competition. He has 8 riddles. Ivory, his friend, has four more riddles than Josh. Taso, another competitor, has twice as many riddles as Ivory. If Josh decides to give away 3 of his riddles and that affects the total number of riddles in the competition, how many riddles does Taso have after accounting for Josh's decision?

**Review Score:** 3,4,4.

**Review Agent Suggestions:**
(1) Clarify the influence of Josh giving away his riddles on Taso's total. For instance, specify that Taso's riddle count is independent and remains unchanged despite Josh's actions. This removes ambiguity regarding the competition's overall structure.
(2) Rephrase the question for better clarity. Instead of saying "and that affects the total number of riddles in the competition," use "Josh's decision to give away his riddles does not change how many riddles Taso has." This direct statement enhances comprehension and ensures readers understand that Taso's initial count remains constant.
(3) Consider providing context as to whether the riddles given away affect only Josh or extend to others. A statement like "regardless of how many riddles Josh gives away, Taso's number of riddles is based on the original count" would maintain logical clarity.

**Revised Problem:** Josh is participating in a school riddle competition. He has 8 riddles. Ivory, his friend, has four more riddles than Josh. Taso, another competitor, has twice as many riddles as Ivory. Josh's decision to give away 3 of his riddles does not change how many riddles Taso has. Regardless of how many riddles Josh gives away, Taso's number of riddles is based on the original count. How many riddles does Taso have?

---

Example of problem refinement by the Reviewer and Revise Agents.

**Problem:** Stella and Twinkle are filling a truck with a capacity of 6000 stone blocks at the rate of 250 blocks per hour per person. They work for four hours without any breaks, after which they are joined by 6 other people who work at the same rate. Calculate the total time taken to fill the truck. Additionally, if the blocks are in the shape of cubes with a side length of 2 units, determine the area of the base of the truck that would hold these blocks when filled to capacity. Provide the total hours taken and the area of the base in square units.

**Rephrased Problem:** Stella and Twinkle are filling a truck with a capacity of 6000 cubic blocks at the rate of 250 blocks per hour per person. They work for four hours without any breaks. After this, they are joined by 6 additional people who work at the same rate. Calculate the total time taken to fill the truck. To begin, calculate the total number of blocks filled by both Stella and Twinkle during their initial 4 hours. Next, determine how long it will take for all workers (Stella, Twinkle, and the 6 additional people) to finish filling the truck. Additionally, the blocks are in the shape of cubes with a side length of 2 units. Calculate the area of the rectangular base of the truck needed to accommodate the cubes when filled to its full capacity, expressed in square units. Provide the total hours taken and the area of the base in square units.

**Review Score:** 4,4,4.

**Review Agent Suggestions:**
(1) Rephrase the initial sentence to: "Stella and Twinkle are filling a truck with a capacity of 6000 cubic blocks, working at a rate of 250 blocks per hour each for four hours without breaks." This enhances clarity by simplifying the phrase structure.
(2) Provide a clearer description of the mathematical operations to carry out after working four hours. For instance, "After 4 hours of work, determine the total blocks filled, and then calculate the time required for the remaining blocks with the help of the additional workers." This improves logical coherence by clarifying the task transition.
(3) Clearly request the output format by stating, "Finally, report both the total hours taken to fill the truck and the area of the rectangular base in square units," ensuring the conclusion is definitive and straightforward for the reader to apprehend.

**Revise Problem:** Stella and Twinkle are filling a truck with a capacity of 6000 cubic blocks, working at a rate of 250 blocks per hour each for four hours without breaks. After 4 hours of work, determine the total blocks filled by both Stella and Twinkle, and then calculate the time required for the remaining blocks with the help of 6 additional workers who work at the same rate. Finally, report both the total hours taken to fill the truck and the area of the rectangular base in square units needed to accommodate the blocks when filled to its full capacity, given that each block is a cube with a side length of 2 units.

---

**Example of problem refinement by the Reviewer and Revise Agents.**

**Problem:** Tom's rabbit can run at a speed of 25 miles per hour, while his cat runs at 20 miles per hour. If the cat gets a 15-minute head start, during which it covers a distance on a flat path, determine the time (in hours) it will take the rabbit to catch up with the cat assuming that for every mile, the rabbit encounters terrain that slows it down by 1 minute for every 5 miles traveled. Additionally, calculate the total distance (in square miles) covered by both animals until the rabbit catches up, assuming their paths can be represented as lines on a coordinate plane.

**Rephrased Problem:** Tom's rabbit can run at a speed of 25 miles per hour, while his cat runs at a speed of 20 miles per hour. The cat gets a 15-minute head start, covering a distance on a flat path during this time. Determine the time (in hours) it will take for the rabbit to catch up to the cat, defined as the moment when both animals have traveled the same linear distance from the starting point. Consider that for every mile the rabbit travels, it encounters terrain that slows its speed by an additional minute for every 5 miles traveled. Finally, calculate the total distance traveled (in miles) by both animals until the rabbit catches up with the cat.

**Review Score:** 3,4,3.

**Review Agent Suggestions:**
(1) Streamline the question for clarity. For example, state: "Tom's rabbit runs at 25 mph and his cat at 20 mph. The cat has a 15-minute head start. When will the rabbit catch up to the cat, considering that every mile the rabbit travels adds an extra minute to its speed?" This revision reduces complexity and provides clearer instructions.
(2) Clarify the impact of the terrain on the rabbit's speed. Instead of stating that its speed will decrease, specify how to adjust the time taken due to the terrain. For instance, rephrase the condition regarding the rabbit's speed to enhance understanding, such as, "For every 5 miles the rabbit runs, add 1 extra minute to its travel time."
(3) Remove unnecessary details and ensure the question leads to a unique solution. Consider separating the catch-up time and total distance calculations into distinct questions or minimizing the effects of terrain to make the problem more straightforward. This way, students can focus on the primary aspects of the problem without layers of complexity that introduce potential for confusion.

**Revised Problem:** Tom's rabbit runs at 25 mph and his cat at 20 mph. The cat has a 15-minute head start. For every 5 miles the rabbit travels, add an extra minute to its travel time. How long (in hours) will it take for the rabbit to catch up to the cat? Also, calculate the total distance (in miles) traveled by both animals when the rabbit catches up.

### A.7.2 EXAMPLES OF PROBLEM REVISIONS FROM MATH

> **Example of problem refinement by the Reviewer and Revise Agents.**
>
> **Problem:** I have 5 marbles numbered 1 through 5 in a bag. Suppose I take out two different marbles at random. What is the expected value of the sum of the numbers on the marbles?
>
> **Rephrased Problem:** You have a bag containing 5 marbles numbered 1 through 5. You randomly draw two different marbles. If marble 1 is drawn first, then the drawn marbles are put back in the bag, and you draw again. If the first marble drawn is not 1, you keep it and draw the second marble without replacement. What is the expected value of the sum of the numbers on the marbles drawn, given that if you draw an odd-numbered marble first, you will always draw marble 2 next, while if you draw an even-numbered marble first, you will draw the next higher number available if possible? Calculate the sum for each scenario and give the overall expected value.
>
> **Review Score:** 3,4,3.
>
> **Review Agent Suggestions:**
> (1) Simplify the language used in the problem statement for better clarity. For instance, instead of saying "if marble 1 is drawn first, then the drawn marbles are put back in the bag," you could say "if marble 1 is drawn first, put the marbles back and draw again." This succinctly conveys the same information without ambiguity.
> (2) Clearly define what happens in all scenarios after each draw. It would be beneficial to write separate scenarios or cases explicitly (e.g., drawing 1, drawing an odd number, drawing an even number), ensuring that all potential outcomes are evident to the reader to aid in understanding and solving the problem.
>
> **Revised Problem:** You have a bag with 5 marbles numbered 1 through 5. You will randomly draw two different marbles according to the following rules: 1. If you draw marble 1 first, put it back in the bag and draw again. 2. If the first marble drawn is not marble 1, you keep that marble and draw a second marble without putting the first back. Additionally: - If the first marble drawn is odd (3 or 5), you will draw marble 2 next. - If the first marble drawn is even (2 or 4), you will draw the next higher number available (for example, if you draw 2 first, you will draw 3 next; if you draw 4, you will draw 5 next). Calculate the expected value of the sum of the numbers on the marbles drawn, considering each scenario.

**Example of problem refinement by the Reviewer and Revise Agents.**

**Problem:** Find the number of ordered pairs of positive integers (a,b) such that a+b=1000 and neither a nor b has a zero digit.

**Rephrased Problem:** In a community project, two teams need to collectively gather exactly 1000 units of resources. Let (a) represent the units collected by Team A and (b) represent those collected by Team B, such that (a + b = 1000). If neither team can collect resources in amounts represented by a number containing the digit zero, how many ordered combinations of (a, b) can the teams realistically plan for their collection, considering all above constraints?

**Review Score:** 4,4,3.

**Review Agent Suggestions:**
(1) Clarify the constraint regarding the digit zero by explicitly stating that both (a) and (b) can only consist of the digits 1-9. This can help eliminate any potential confusion about what is meant by not containing the digit zero.
(2) Define what is meant by "ordered combinations". It could be interpreted in different ways—does it mean different pairings of (a) and (b) that total 1000, or is it simply asking for the number of distinct pairs? Providing examples or additional context could further enhance comprehension.
(3) Instead of stating "the teams realistically plan for their collection," consider specifying that this refers to the mathematical combinations that satisfy all conditions. This would reinforce the focus of the question on mathematics rather than a broader planning context.

**Revised Problem:** In a community project, two teams need to collectively gather exactly 1000 units of resources. Let (a) represent the units collected by Team A and (b) represent those collected by Team B, such that (a + b = 1000). Both (a) and (b) must be composed exclusively of the digits 1 through 9, meaning neither can include the digit zero. How many distinct ordered pairs ((a, b)) satisfy this condition, ensuring all constraints are met?

## A.8   AGENT PROMPT

---

**Example 1: Problem Rating Prompt**

As a mathematics quality evaluator, your task is to rigorously assess whether a given mathematical question is high-quality and valuable as a reference for synthesizing new questions. Use the following criteria:

1. Complexity (1–10): How many logical steps or conceptual layers are required? Does it integrate multiple mathematical domains (e.g., algebra + geometry) or demand critical thinking?
2. Information Value: Does it contain useful knowledge or reasoning opportunities? Can it help learners discover concepts, strategies, or patterns?
3. Clarity & Precision (1–10): Is the question unambiguous, logically consistent, and free of errors? Poorly framed questions score lower.

** Scoring Guidelines **:
- Please rate the sample on a scale from 1 to 10 for each criterion, and return an overall rating on a scale from 1 to 10, where a higher score indicates higher level of quality.
- Ensure that the ratings are not overly concentrated around a specific score. If multiple samples have similar qualities, consider spreading the scores more evenly to reflect subtle differences.
- Penalize heavily for ambiguity, errors, or oversimplification.

Please carefully evaluate the following data sample and return the integral evaluation scores using the JSON format:
{
"Complexity": <number, 1–10>,
"Information Value": <number, 1–10>,
"Clarity": <number, 1–10>,
"Overall rating": <number, 1–10>
}

---

**Example 2: Problem Rephrase Prompt**

Act as an expert mathematics educator specializing in problem complexity escalation. Systematically transform the given problem while preserving its core concepts, using the following framework:

**Stage 1: Problem Deconstruction**
- Domain Identification: [Algebra/Geometry/Calculus/etc.]
- Core Competencies: [List specific theorems/formulas/methods]
- Baseline Difficulty: [Level 1–5 using Krathwohl's Cognitive Rigor Index]

**Stage 2: Escalation Protocol**
Select $\geq 3$ complexity dimensions from:
1. Multi-stage Transformation: Designs a single, cohesive mathematical problem where the complete solution inherently demands multiple, sequentially dependent calculations. The output of one implicit intermediate step must serve as the essential and sole input for the next, creating a longer chain of necessary computational derivation for the solver to reach the definite final answer.
2. Cross-domain Integration: Create hybrid problems combining $\geq 2$ mathematical disciplines
3. Real-world Parameterization: Embed contextual constraints with multivariate relationships
4. Conditional Branching: Introduce layered constraints requiring decision-tree analysis
5. Inverse Problem Design: Reverse-engineer given solutions to reconstruct premises
6. Uncertainty Integration: Incorporate measurement errors/probabilistic factors
7. Optimization Extension: Convert closed solutions into multi-objective optimization challenges

**Stage 3: Revise question**
- Must be a definitive mathematical problem: The question must require mathematical reasoning, calculation, or logical deduction.
- Must have a unique and specific mathematical answer: The problem should lead to a single, verifiable numerical or analytical solution, avoiding open-ended questions, subjective evaluations, or non-mathematical tasks.

Please reply strictly in the following format:
Stage 1
#Problem Deconstruction#:
Stage 2
#Escalation Protocol#:
Stage 3
#Finally Rewritten question#:

**Example 3: Problem Review Prompt**

As a mathematics quality checker, your task is to rigorously assess whether a given mathematical question is high-quality and provide rewrite suggestions:

1. Clarity & Grammar (1–5): The question must be grammatically correct, precisely phrased, and easy to understand. It should avoid ambiguity in wording or phrasing.

2. Logical Coherence & Completeness (1–5): All elements of the problem (e.g., given information, constraints, relationships, objectives) must be logically interconnected and sufficient. The problem should present a clear, sequential path for reasoning, without missing information required for the specified solution approach.

3. Mathematical Validity & Solvability (1–5): The problem must be fundamentally a mathematics problem, with all its premises and conditions being *mutually consistent* and *mathematically sound*. It must lead to a *unique, solvable numerical or analytical answer* that adheres to all mathematical rules and specified ranges (e.g., probabilities summing to 1, valid geometric properties, real number solutions). If any condition leads to a mathematical contradiction or an impossible/undefined solution (e.g., total probability > 1 after adjustments, an equation with no valid solution within given constraints), this criterion rates very low, and the exact mathematical inconsistency must be pinpointed. Avoid open-ended or non-mathematical questions.

** Scoring Guidelines **:

- Please rate the sample on a scale from 1 to 5 for each criterion, and return an overall rating on a scale from 1 to 5, where a higher score indicates higher level of quality.

Rephrased question: {rephrased_question}

**Output Requirements**

Respond in the following plain-text format **only** (do not include JSON or any additional commentary):

###thought###

<Analytical reasoning addressing each criterion sequentially, especially for rephrased_question >

###rating_score###

<Clarity & Grammar score >, <Logical Consistency score >, <Mathematical Relevance & Solvability score >

###suggestions###

###Specific improvement 1###

<Specific improvement 1 >

###Specific improvement 2###

<Specific improvement 2 >

...more improvements if needed...

Noice:

- "rating_score" represents evaluate score of Rephrased question.

- when generate "suggestions", please give more details and reasons for each improvement.

**Example 4: Problem Revise Prompt**

As an expert in mathematical question improvement, please optimize the question according to the following suggestions:
{suggestions}

Optimization requirements:
1. Clarity & Grammar (1–5): The question must be grammatically correct, precisely phrased, and easy to understand. It should avoid ambiguity in wording or phrasing.
2. Logical Coherence & Completeness (1–5): All elements of the problem (e.g., given information, constraints, relationships, objectives) must be logically interconnected and sufficient. The problem should present a clear, sequential path for reasoning, without missing information required for the specified solution approach.
3. Mathematical Validity & Solvability (1–5): The problem must be fundamentally a mathematics problem, with all its premises and conditions being *mutually consistent* and *mathematically sound*. It must lead to a *unique, solvable numerical or analytical answer* that adheres to all mathematical rules and specified ranges (e.g., probabilities summing to 1, valid geometric properties, real number solutions). If any condition leads to a mathematical contradiction or an impossible/undefined solution (e.g., total probability exceeds 1 after adjustments, an equation with no valid solution within given constraints), this criterion rates very low, and the exact mathematical inconsistency must be pinpointed. Avoid open-ended or non-mathematical questions.
original question: {rephrased_question}

** Output Requirements **
Respond in the following plain-text format **only** (do not include JSON or any additional commentary):
###revised_question###
<improved full question>
###revision_notes###
<Specific revision note>

**Example 5: Solution Generation Prompt (GSM8K)**

As a mathematics problem solving expert, analyze and answer the following question.

Workflow:
1. Analyze and Deconstruct:
- First, systematically break down the problem into its core components.
- Explicitly list all given data, variables, constraints, and the final objective of the problem.
2. Clarify Ambiguities:
- Before starting calculations, if any part of the problem statement is ambiguous, you must state your interpretation and the reasoning behind it.
3. Step-by-Step Derivation and Process Demonstration:
- For each component of the problem, provide a detailed step-by-step derivation.
- You must show all intermediate calculation steps, formulas used, and logical judgments. Do not skip or summarize critical calculation processes.
- For any step involving complex calculations, multi-case analysis, or iterative enumeration (e.g., filtering combinations that meet a condition, solving systems of equations, analyzing multiple scenarios), you must clearly list all cases or combinations considered.
4. Synthesis and Final Calculation:
- Integrate the results from all preceding steps to perform the final calculation.
- Clearly show the final calculation that leads to the final answer.

Respond in the following plain-text format **only** (do not include JSON or any additional commentary):
###thought### <step-by-step reasoning process> ###answer### <final answer>

Output Notice:
- Replace <step-by-step reasoning process> with your detailed derivation.
- Replace <final answer> with the concise final answer (e.g., a number or fraction), without units or extra words.

Output Example 1:

Question: A cleaning company produces two sanitizer sprays. One spray kills 50% of germs, and another spray kills 25% of germs. However, 5% of the germs they kill are the same ones. What percentage of germs would be left after using both sanitizer sprays together?

Output(must match the specified format exactly):
###thought### To correctly calculate the percentage of germs left, we must use the Principle of Inclusion-Exclusion to find the total percentage of unique germs killed ......
###answer### 30

Question: {question}

Output:

**Example 6: Solution Generation Prompt (MATH)**

As a mathematics problem solving expert, analyze and answer the following question.

Workflow:
1. Analyze and Deconstruct:
- First, systematically break down the problem into its core components.
- Explicitly list all given data, variables, constraints, and the final objective of the problem.
2. Clarify Ambiguities:
- Before starting calculations, if any part of the problem statement is ambiguous, you must state your interpretation and the reasoning behind it.
3. Step-by-Step Derivation and Process Demonstration:
- For each component of the problem, provide a detailed step-by-step derivation.
- You must show all intermediate calculation steps, formulas used, and logical judgments. Do not skip or summarize critical calculation processes.
- For any step involving complex calculations, multi-case analysis, or iterative enumeration (e.g., filtering combinations that meet a condition, solving systems of equations, analyzing multiple scenarios), you must clearly list all cases or combinations considered.
4. Synthesis and Final Calculation:
- Integrate the results from all preceding steps to perform the final calculation.
- Clearly show the final calculation that leads to the final answer.

Respond in the following plain-text format **only** (do not include JSON or any additional commentary):
###thought### <step-by-step reasoning process> ###answer### <final answer>

Output Notice:
- Replace <step-by-step reasoning process> with your detailed derivation.
- Replace <final answer> with the concise final answer (e.g., a number or fraction), without units or extra words.

Output Example 1:

Question: A box contains 5 white balls and 6 black balls. Two balls are drawn out of the box at random. What is the probability that they both are white?

Output(must match the specified format exactly):
###thought### To solve for the probability of drawing two white balls from a box containing 5 white and 6 black balls, we'll use......
###answer### $\dfrac{2}{11}$

Question: {question}

Output:

