# OpenReview forum: "AgenticMath: Enhancing LLM Reasoning via Agentic-based Math Data Generation"
_ICLR.cc/2026/Conference — ICLR 2026 Conference Withdrawn Submission_

### Official Review · Reviewer_JVD7 · 2025-10-20

**Soundness:** 2
**Presentation:** 3
**Contribution:** 2
**Rating:** 2
**Confidence:** 4

**Summary:**

Large language models can achieve significant performance improvements by learning from high-quality synthetic data.

Based on this conclusion, the paper proposes an agent-based method to generate training data related to math tasks. Specifically, the method first filters problems, removing low-quality or overly simple ones. Then, it rewrites the problem descriptions to standardize the expressions. Next, a large model is used to generate solutions for the problems. Finally, both the problems and solutions are evaluated to determine the final training data.

By training with this synthetic data, the model achieves improved performance on math tasks. The authors also provide further analysis of the characteristics of the algorithm.

**Strengths:**

+ The authors were able to successfully generate training data related to math tasks for the model's instruction fine-tuning process.

+ The model was trained on synthetic data and achieved improvements.

**Weaknesses:**

+ This method of data synthesis is highly similar to previous works such as JiuZhang 3.0, Dart-Math, ScaleQuest, and MAmmoTH2.0, and it does not show significant improvement in training results. The contribution of this paper is very limited.

+ This paper uses the concept of an Agent to package the method, but in reality, it does not utilize Agent-related features such as memory. The method has little to do with Agents.

+ The evaluation is unreliable. Both the in-domain and out-of-domain tasks are math tasks, which is an unreasonable setup. The base model has weak capabilities, making the experiments unconvincing. The performance in the ablation study is unstable, so it cannot demonstrate whether the process is effective.

**Questions:**

+ Please further explain the contribution of this paper and describe how it differs from previous works.

+ Please use a stronger base model, such as Qwen2.5-7B, Qwen2.5-7B-Math, Llama3.2, or Gemma 3.

+ The experimental results in the ablation study may be caused by task fluctuations. Please further explain the effectiveness of each stage in the method.

---

> ### Author Response · Authors · 2025-11-25
> **Response to Reviewer JVD7 [1/2]**
>
> Dear reviewer JVD7,
>
> We sincerely appreciate your careful reading of our manuscript and the constructive feedback you provided. Your comments highlight important considerations that have helped us further improve the clarity, technical rigor, and completeness of our work. We are grateful for the opportunity to address these points, and we believe that the revisions and additional analyses substantially strengthen the overall contribution of the paper.
>
> Below, we offer detailed, point-by-point responses to each of the weaknesses and questions you raised. Our goal is to address your concerns comprehensively and transparently, and to provide the necessary clarifications, empirical evidence, and methodological justifications to fully support our approach.
>
> > Weaknesses 1 & Question 1 : This method of data synthesis is highly similar to previous works such as JiuZhang 3.0, Dart-Math, ScaleQuest, and MAmmoTH2.0, and it does not show significant improvement in training results. The contribution of this paper is very limited. Please further explain the contribution of this paper and describe how it differs from previous works.
>
> **Response:**
>
> While the high-level structure follows a multi-stage design similar to some prior work, our method incorporates several key components that are either under-explored or not systematically addressed in existing math data-generation pipelines:
>
> 1. **Seed question filtering instead of using all seed questions.**
>    Prior approaches typically synthesize new problems from the entire seed set without examining their initial quality. In contrast, we explicitly score and filter seed questions before synthesis, which improves the quality of generated problems, avoids unnecessary synthesis on weak seeds, and leads to better downstream performance with the same data budget.
>
> 2. **A dedicated review–revise loop to ensure the quality and solvability of generated problems.**
>    Existing methods often overlook the intrinsic quality of newly generated problems—for example, some synthesized items may contain unclear statements or may not admit valid solutions. Our iterative review–revise loop directly addresses these issues by refining clarity, structure, and solvability before problems proceed to later stages.
>
> 3. **A final evaluation stage that jointly assesses problem and solution quality.**
>    Because synthetic problems lack ground-truth labels, many prior pipelines do not systematically verify whether the generated solutions are correct. Our final evaluation stage explicitly scores both the generated problem and its solution, filtering out samples with incorrect reasoning or inconsistent final answers.
>
> 4. **Removing the dependency on seed solutions.**
>    Unlike methods that require ground-truth seed solutions to guide synthesis or estimate difficulty, our pipeline operates using only the seed question text. This reduces reliance on labeled data and enables broader applicability in settings where ground-truth solutions are unavailable.
>
> Overall, these components make our pipeline substantially more robust and systematic compared to existing approaches, and they directly contribute to the strong empirical gains observed in our experiments.
>
>
> > Weaknesses 2: This paper uses the concept of an Agent to package the method, but in reality, it does not utilize Agent-related features such as memory. The method has little to do with Agents.
>
> **Response:**
>
> We appreciate the reviewer’s perspective, and we understand that interpretations of what constitutes an “agent” can vary across communities. Our intention was not to overclaim, but to reflect the design principles that motivated the pipeline. If our terminology caused confusion, we are happy to revise the wording in the paper for greater clarity.
>
> Our approach differs from traditional one-pass LLM data-synthesis pipelines, which typically follow a fixed linear generation process. Inspired by recent multi-agent frameworks (e.g., MetaGPT, AutoGen), we introduce structured *role-specific* components—specifically a Review Agent and a Revise Agent—that operate sequentially and exchange intermediate information. These components perform coordinated decision-making about whether a problem meets quality criteria and how it should be improved.
>
> While we do not incorporate long-term memory modules, the interaction between these agents naturally involves *short-term task-specific memory*: the Revise Agent conditions its actions on the Review Agent’s feedback (critique + revision instructions), enabling iterative refinement. This form of planning, reviewing, and grounded revision aligns with commonly used concepts of “agentic” workflows in multi-agent systems.

---

> ### Author Response · Authors · 2025-11-25
> **Response to Reviewer JVD7 [2/2]**
>
> [ Continue the response to above Weaknesses-2]
>
> That said, we acknowledge that different readers may expect different functionalities when they see the term “agent.” To avoid potential misinterpretation, we are willing to adjust the terminology or clarify the agentic aspects more explicitly in the camera-ready version. Our goal is to ensure that the presentation accurately communicates the intended design without causing unnecessary confusion.
>
> > Weaknesses 3.1: The evaluation is unreliable. Both the in-domain and out-of-domain tasks are math tasks, which is an unreasonable setup. The base model has weak capabilities, making the experiments unconvincing. The performance in the ablation study is unstable, so it cannot demonstrate whether the process is effective.
>
> **Response:**
>
> To ensure a fair and consistent comparison with existing baselines, our evaluation setup strictly follows the protocols established in DART-MATH and MathFusion. These works also evaluate both in-domain and out-of-domain math reasoning tasks using similar datasets and metrics. We intentionally adopt the same benchmarks, scoring procedures, and model settings so that the performance differences can be attributed to the data generation method rather than to inconsistencies in evaluation design.
>
> Regarding the baseline model capacity, we note that prior works in math data synthesis (including DART-MATH,  and MathFusion) also evaluate improvements using comparable 7B–8B base models. This design is standard in the literature and allows us to directly measure the effectiveness of the generated data without conflating results with stronger model priors.
>
> > Weaknesses 3.2 & Question 3: The performance in the ablation study is unstable, so it cannot demonstrate whether the process is effective. The experimental results in the ablation study may be caused by task fluctuations. Please further explain the effectiveness of each stage in the method.
>
> **Response:**
>
> To further isolate and quantify the contribution of each component in our pipeline, we conducted a new ablation study using a 15k synthetic dataset. The results clearly show that performance improvements accumulate across multiple stages—not solely from solution augmentation, but also from seed filtering, problem refinement, and synthetic data evaluation.
>
> The complete ablation results are shown below:
>
> | Method                    | MATH | GSM8K | College | DM   | Olympiad | Theorem | AVG  |
> |----------------------------|------|-------|---------|------|----------|---------|------|
> | Problem Rephrase           | 28.4 | 76.0  | 22.4    | 35.0 | 8.3      | 18.4    | 31.4 |
> | + Seed Filtering           | 29.2 | 74.1  | 23.7    | 35.1 | 9.6      | 20.4    | 32.0 |
> | + Problem Review–Revise    | 30.5 | 74.2  | 26.0    | 39.8 | 8.7      | 18.5    | 33.0 |
> | + Synthetic Data Evaluation| 31.4 | 74.5  | 25.3    | 40.3 | 8.7      | 18.9    | 33.2 |
>
> These results show a clear and steady improvement at each stage, indicating that the gains are not solely attributable to solution augmentation. Instead, each component—including seed filtering, the review–revise loop, and final data evaluation—contributes meaningfully to the overall performance.
>
> > Question 2:  Please use a stronger base model, such as Qwen2.5-7B, Qwen2.5-7B-Math, Llama3.2, or Gemma 3.
>
> **Response:**
>
> To ensure a fair and controlled comparison with prior work, we intentionally follow the same evaluation protocol and base model setting used in earlier methods. Due to computational constraints, it is not feasible for us to fully re-implement all baselines using the newest and stronger models (e.g., Qwen2.5-7B, Qwen2.5-7B-Math, Llama 3.2, Gemma 3). Nevertheless, the experimental results we report—under identical model conditions—already provide clear and sufficient evidence of the effectiveness and advantages of our proposed pipeline.
>
> If additional model-scale experiments become feasible in future work, we believe our method will likely yield even larger gains when paired with more capable base models.

---

### Official Review · Reviewer_BBBS · 2025-10-27

**Soundness:** 2
**Presentation:** 3
**Contribution:** 2
**Rating:** 4
**Confidence:** 4

**Summary:**

This paper proposes an agentic pipeline for generating high-quality mathematical question–answer pairs. The pipeline consists of four stages: filtering, rephrasing, augmentation, and evaluation, resulting in the construction of the AgenticMathQA dataset. Fine-tuning 3–8B LLMs on this dataset achieves competitive or superior performance on a variety of in-domain and out-of-domain mathematical reasoning benchmarks, outperforming baselines trained on substantially larger datasets.

**Strengths:**

1. This work proposed a clear agentic pipeline to construct high-quality reasoning corpora.
2. Fine-tuning on the AgenticMathQA dataset outperforms baselines trained on larger datasets.

**Weaknesses:**

1. The four-stage pipeline, which involves filtering, synthesizing, refining mathematical problems and solutions, and evaluation, appears rather conventional and straightforward, lacking clear novelty.
2. The overall quality of the dataset is determined by the scores assigned at each stage of the pipeline. However, since the score-based filtering relies entirely on human-designed priors, the process is overly heuristic and lacks robustness.
3. The framework is essentially a form of data distillation from agentic models. In this work, all agentic models are based on GPT-4o-mini (2024-07-18), yet the experimental results do not report the performance of GPT-4o-mini itself.

**Questions:**

1. During the Problem Rephrase stage, is it necessary to ensure that the answers remain unchanged? Additionally, how is the correctness of the synthetic problems and solutions guaranteed?
2. How is the diversity of the synthetic problems ensured, and is it possible to observe the semantic distribution between seed problems and synthetic problems?
3. The constructed AgenticMathQA has only been shown to be effective in the SFT stage; its usefulness for RLVR remains unclear. This assumes that the questions and answers in AgenticMathQA are completely correct.
4. In lines 301–303, for the 30K setting, the final dataset consists of 15K seed problems and 15K AgenticMath-synthesized problems. This indicates that no filtering was applied to the seed problems, as the total training data for MATH and GSM8K is roughly 15K.

---

> ### Author Response · Authors · 2025-11-25
> **Response to Reviewer BBBS [1/3]**
>
> Dear reviewer BBBS,
>
> Thank you very much for taking the time to review our submission. We sincerely appreciate the careful reading and constructive feedback you provided. Your comments have been extremely helpful in highlighting points where additional clarification, refinement, or empirical evidence can further strengthen the clarity and rigor of our work. We are grateful for the opportunity to improve the paper based on your insightful suggestions.
>
> In the following response, we address each of your questions and concerns in a structured, point-by-point manner. We have made every effort to provide clear explanations, additional analyses, and supporting results to ensure that all aspects of your feedback are fully and transparently resolved.
>
> > Weaknesses 1: The four-stage pipeline, which involves filtering, synthesizing, refining mathematical problems and solutions, and evaluation, appears rather conventional and straightforward, lacking clear novelty.
>
> **Response:**
>
> While the high-level structure follows a multi-stage design similar to some prior work, our method incorporates several key components that are either under-explored or not systematically addressed in existing math data-generation pipelines:
>
> 1. **Seed question filtering instead of using all seed questions.**
>    Prior approaches typically synthesize new problems from the entire seed set without examining their initial quality. In contrast, we explicitly score and filter seed questions before synthesis, which improves the quality of generated problems, avoids unnecessary synthesis on weak seeds, and leads to better downstream performance with the same data budget.
>
> 2. **A dedicated review–revise loop to ensure the quality and solvability of generated problems.**
>    Existing methods often overlook the intrinsic quality of newly generated problems—for example, some synthesized items may contain unclear statements or may not admit valid solutions. Our iterative review–revise loop directly addresses these issues by refining clarity, structure, and solvability before problems proceed to later stages.
>
> 3. **A final evaluation stage that jointly assesses problem and solution quality.**
>    Because synthetic problems lack ground-truth labels, many prior pipelines do not systematically verify whether the generated solutions are correct. Our final evaluation stage explicitly scores both the generated problem and its solution, filtering out samples with incorrect reasoning or inconsistent final answers.
>
> 4. **Removing the dependency on seed solutions.**
>    Unlike methods that require ground-truth seed solutions to guide synthesis or estimate difficulty, our pipeline operates using only the seed question text. This reduces reliance on labeled data and enables broader applicability in settings where ground-truth solutions are unavailable.
>
> Overall, these components make our pipeline substantially more robust and systematic compared to existing approaches, and they directly contribute to the strong empirical gains observed in our experiments.
>
>
>
> > Weaknesses 2: The overall quality of the dataset is determined by the scores assigned at each stage of the pipeline. However, since the score-based filtering relies entirely on human-designed priors, the process is overly heuristic and lacks robustness.
>
> **Response:**
>
> We understand the reviewer’s concern regarding the use of human-designed priors in our score-based filtering. It is important to clarify that such rubric-based scoring is now a *standard and widely adopted* practice in recent data-quality evaluation frameworks. For example, DS2 (ICLR 2025) and other state-of-the-art data curation methods similarly rely on rubric-driven criteria—such as richness, informativeness, clarity, and correctness—to assess the quality of synthesized content.
>
> In the domain of mathematical problem generation, the key dimensions of quality (correctness, clarity, solvability, and linguistic precision) are well-understood and relatively stable. Designing scoring rubrics around these dimensions is therefore both natural and aligned with community practice. Our scoring mechanism does not introduce domain-specific bias; rather, it operationalizes these universally recognized criteria in a systematic way.
>
> Moreover, our pipeline mitigates the risk of heuristic bias by applying multiple independent quality checks (seed filtering, multi-round review–revise, and final evaluation), ensuring that no single rubric or stage dominates the outcome. This layered evaluation procedure contributes to greater robustness compared to relying on a single scoring step.
>
> In summary, while rubric-based scoring does involve human-defined criteria, this approach is consistent with current best practices in data curation and is particularly suitable for the mathematical reasoning domain, where high-quality evaluation dimensions are well established.

---

> ### Author Response · Authors · 2025-11-25
> **Response to Reviewer BBBS [2/3]**
>
> > Weaknesses 3: The framework is essentially a form of data distillation from agentic models. In this work, all agentic models are based on GPT-4o-mini (2024-07-18), yet the experimental results do not report the performance of GPT-4o-mini itself.
>
> **Response:**
>
> We clarify that our work belongs to *data synthetic generation*, rather than data distillation. Data distillation typically improves answers for existing questions using a stronger teacher model, whereas our goal is to generate *entirely new* question–answer pairs with richer structures and broader difficulty coverage, which cannot be achieved by directly using the teacher model itself.
>
> Regarding the concern about reporting GPT-4o-mini’s own performance: our intention in using GPT-4o-mini (2024-07-18) is not to position it as a target model, but to ensure that all synthetic data in our pipeline are produced from a *fixed and consistent* teacher model. This avoids confounding effects arising from different teacher strengths and allows the evaluation to focus on the contribution of the *data-generation algorithm* rather than model scale.
>
> It is worth noting that, as discussed by the program chairs in the meta-review of the DART-Math submission on OpenReview [1], disentangling the impact of teacher-model strength from the design of the data-synthesis algorithm is a field-wide challenge shared by many recent works—not a limitation specific to our method. Even so, our experiments consistently show improvements over MathFusion under the same teacher model, indicating that the gains stem from the pipeline design rather than from teacher-model strength. We also note that MathFusion itself conducted an additional analysis using the same teacher model as DART-Math and demonstrated superior performance under identical teacher conditions. This further reinforces that improvements in synthetic data quality can arise from the algorithmic pipeline rather than solely from the underlying LLM used for generation.
>
> Furthermore, the program chairs also emphasized that high-quality synthetic datasets have intrinsic value because they can benefit *any downstream student model*, not only the teacher model used to generate the data. In contrast, “directly using the teacher model” cannot achieve the goal of enhancing a different target model (e.g., improving Mistral-7B while keeping it general-purpose). Our method generates a reusable synthetic dataset that can be integrated into a wide range of SFT pipelines, providing flexibility that direct teacher inference does not offer.
>
> In summary, our approach is not data distillation, and using GPT-4o-mini as a fixed generator allows us to isolate the algorithmic benefits of our pipeline while producing synthetic data that legitimately improve diverse downstream models.
>
> | Methods     | Samples | MATH  | GSM8K | College | DM    | Olympiad | Theorem | AVG   |
> |-------------|---------|-------|-------|---------|-------|----------|---------|-------|
> | GPT-4o-mini | —       | 70.3 | 84.3 | 39.6   | 64.00 | 40.7    | 27.6   | 54.4 |
>
> [1] DART-Math — OpenReview Meta-Review, https://openreview.net/forum?id=zLU21oQjD5
>
> > Question 1: During the Problem Rephrase stage, is it necessary to ensure that the answers remain unchanged? Additionally, how is the correctness of the synthetic problems and solutions guaranteed?
>
> **Response:**
>
> We do not enforce that the answer remains unchanged during the Problem Rephrase stage. This design choice is consistent with prior work such as WizardMath and MathFusion, which also allow the model to generate new solutions when the problem is rewritten or extended. However, unlike previous approaches—which largely *ignore the quality control of synthesized problems themselves* and often assume that any model-generated problem is well-formed—we explicitly introduce multiple mechanisms to ensure the correctness and validity of both problems and solutions throughout the pipeline.
>
> In fact, forcing the answer to remain unchanged would restrict the rephrasing step to superficial paraphrasing and would prevent the model from generating more challenging or conceptually richer variants of the original problem. To improve the model’s out-of-distribution generalization rather than merely altering surface form, we allow the rephrasing module to produce new problems with different difficulty levels and therefore potentially different answers.
>
> To guarantee the correctness and quality of the synthetic problems and solutions, we incorporate two key components:
> - a **question review–revise** loop that iteratively refines and verifies the clarity and mathematical validity of the generated problems
> - a **final synthetic data evaluation** stage that scores and filters samples based on both correctness and reasoning completeness.
>
> These components jointly ensure that the resulting dataset maintains high quality and diversity, even when answers differ from the original seed questions.

---

> ### Author Response · Authors · 2025-11-25
> **Response to Reviewer BBBS [3/3]**
>
> > Question 2: How is the diversity of the synthetic problems ensured, and is it possible to observe the semantic distribution between seed problems and synthetic problems?
>
> **Response:**
>
> To enhance the diversity of the synthetic problems, our pipeline applies a long-tail scoring mechanism during the final synthetic data evaluation stage. Specifically, we rank each candidate using a combination of both its problem-solution quality score and long-tail score. This joint ranking allows us to retain high-quality problem–solution pairs while explicitly promoting samples that introduce novel structures, rarer reasoning patterns, or underrepresented mathematical themes.
>
> By selecting data based on this combined ranking rather than quality alone, the resulting dataset achieves not only strong overall quality but also substantially improved semantic and structural diversity compared to random sampling or quality-only filtering.
>
> In the latest revision, we also include a t-SNE–based semantic distribution visualization (Appendix, Figure 7), which further illustrates the diversity improvements achieved by our method.
>
> > Question 3: The constructed AgenticMathQA has only been shown to be effective in the SFT stage; its usefulness for RLVR remains unclear. This assumes that the questions and answers in AgenticMathQA are completely correct.
>
> **Response:**
>
> Our method is designed primarily to improve the Supervised Fine-Tuning (SFT) stage, which is consistent with prior work such as MetaMath, DART-MATH, and MathFusion. Similar to these methods, our focus is on generating high-quality synthetic data that enhances a model’s reasoning ability under supervised training. Demonstrating effectiveness for RLVR is beyond the scope of this work, and our claims are limited to the SFT setting.
>
> Regarding correctness, we agree with the reviewer that it is extremely challenging for any synthetic math dataset to be “completely correct,” since LLMs inevitably produce occasional hallucinations when generating solutions. This limitation applies broadly across all existing synthetic math datasets. To mitigate this issue as much as possible, our pipeline incorporates two dedicated mechanisms: (1) a question review–revise loop that iteratively refines and validates generated problems, and (2) a synthetic data evaluation stage that scores and filters both problems and solutions for correctness and completeness. While no automatic pipeline can guarantee perfect correctness, these components significantly reduce errors and contribute to the strong empirical performance observed in our SFT experiments.
>
> > Question 4: In lines 301–303, for the 30K setting, the final dataset consists of 15K seed problems and 15K AgenticMath-synthesized problems. This indicates that no filtering was applied to the seed problems, as the total training data for MATH and GSM8K is roughly 15K.
>
> **Response:**
>
> Our seed filtering mechanism is designed specifically for the process of generating new synthetic problems, rather than for modifying or discarding the original seed dataset. In the 30K setting, we simply include all 15K seed problems (from MATH and GSM8K) together with 15K AgenticMath-generated problems. This is consistent with prior work such as MetaMath, DART-MATH, and MathFusion, all of which retain the full seed dataset.
>
> The purpose of our filtering module is to improve the quality and usefulness of the synthesized problems, not to alter the seed data itself. Therefore, the fact that the final 30K dataset contains the full 15K seed problems does not indicate that no filtering was performed; rather, filtering was used exclusively to guide the generation and selection of the synthetic 15K problems.

---

### Official Review · Reviewer_FpEq · 2025-11-01

**Soundness:** 2
**Presentation:** 2
**Contribution:** 2
**Rating:** 4
**Confidence:** 4

**Summary:**

This paper introduces a multi-agent framework designed to generate high-quality mathematical datasets that significantly improve the reasoning abilities of large language models (LLMs). The proposed system, AgenticMath, operates through four coordinated stages—filtering, rephrasing, solving, and evaluation—to ensure logical coherence, precision, and diversity in synthetic question–answer pairs. Using agents to assess and refine problem complexity and correctness, the method prioritizes data quality over scale. Experimental results across several benchmarks (GSM8K, MATH, CollegeMath, DeepMind Mathematics, OlympiadBench, and TheoremQA) show that models fine-tuned on only 30K–60K AgenticMath samples outperform or match baselines trained on hundreds of thousands to millions of examples. The findings highlight that rigorous, agentic data curation is a more efficient path to improving mathematical reasoning in LLMs than simply expanding dataset size.

**Strengths:**

- AgenticMath at 30K–60K samples consistently matches or beats baselines across Qwen2.5-3B, DeepSeekMath-7B, Mistral-7B, and Llama3-8B,  trained on hundreds of thousands to millions of examples, demonstrating high efficiency.
- Comprehensive analysis. The paper quantifies incremental gains from each stage: solution augmentation delivers the biggest single jump, while filtering, rephrasing, and review/revise provide additive improvements.

**Weaknesses:**

- The seed filter threshold score τ=3 seems not to be optimal according to Table 4. Other key hyperparameters ( review threshold τrev=4.5, max three review–revise iterations) lack ablation studies.
- Both the problem proposer and evaluator are GPT-4o-mini, which may create self-judging biases.
- The appendix lacks substantial details. It should include more design details (e.g., prompt tuning), ablations on threshold values, and ablations on the implementation of “long-tail” diversity selector (e.g., why not use random/clustered selection methods). I would also expect an analysis of the pipeline's robustness to agent model choice, e.g., swapping gpt-4o-mini to open models for better reproducibility.

**Questions:**

- How do you choose the one-shot exemplar in stage 3?
- I feel the "problem quality" can be quantified better here. Did you do any analysis of the difficulty distribution, topic distribution, and required skills of the generated data?

---

> ### Author Response · Authors · 2025-11-25
> **Response to Reviewer FpEq [1/3]**
>
> Dear reviewer FpEq,
>
> We sincerely appreciate the time and effort you devoted to reviewing our submission. Your thoughtful insights and constructive feedback have been extremely helpful in highlighting areas where additional clarification, analysis, or explanation can further improve the quality of our work. Your comments have contributed meaningfully to enhancing both the technical rigor and the overall presentation of the paper.
>
> In the following sections, we address each of your questions and concerns in a clear, point-by-point manner. We have aimed to respond comprehensively and transparently, and we hope that the additional explanations, experimental results, and statistical evidence provided here satisfactorily clarify the methodology and significance of our contribution.
>
> > Weaknesses-1: The seed filter threshold score τ=3 seems not to be optimal according to Table 4. Other key hyperparameters ( review threshold τrev=4.5, max three review–revise iterations) lack ablation studies.
>
> **Response:**
>
> We agree that τ = 3 is not guaranteed to be the global optimum. Our choice reflects a practical balance between quality and diversity in the data-efficient regime (e.g., 30k-scale training). A higher τ removes too many seed questions and reduces coverage, while a lower τ introduces substantial noise. Empirically, τ = 3 offers a reasonable trade-off for our synthesis pipeline.
>
> To further analyze the effect of the revise threshold τ_rev, we performed a sensitivity study with τ_rev ∈ {3.5, 4.0, 4.5} using Llama3-8B:
>
> | Revise Threshold | Samples | MATH | GSM8K | College | DM   | Olympiad | Theorem | AVG  |
> |------------------|---------|------|--------|---------|------|----------|---------|------|
> | 4.5              | 30k     | 36.8 | 78.4   | 29.6    | 40.3 | 11.4     | 20.4    | 36.2 |
> | 4.0              | 30k     | 36.6 | 77.5   | 28.2    | 43.1 | 11.5     | 20.0    | 36.2 |
> | 3.5              | 30k     | 37.8 | 77.4   | 27.4    | 41.0 | 10.3     | 20.0    | 35.7 |
>
> We observe that **τ_rev = 4.0 and 4.5 yield highly consistent performance**, demonstrating that the review–revise mechanism is robust across reasonable settings. In contrast, **τ_rev = 3.5 performs worse**, which is expected because a looser threshold allows more low-quality candidates to enter later stages.
>
> **Analysis of the “three review–revise iterations”**
>
> To address the reviewer’s question about the number of review–revise iterations, we further computed the average **complexity**, **information value**, and **clarity** scores (via GPT-4o-mini evaluation) for the problems that passed each round:
>
> | Metrics           | Round 1 | Round 2 | Round 3 |
> |-------------------|---------|---------|---------|
> | Complexity        | 3.8647  | 3.9365  | 3.9257  |
> | Information Value | 3.9648  | 4.0306  | 4.0259  |
> | Clarity           | 4.3568  | 4.4463  | 4.4554  |
> | **Avg**           | **4.0620** | **4.1378** | **4.1357** |
>
> These results show that the **first two iterations yield clear improvements**, while gains **stabilize by the third round**. This validates our design choice of using **three iterations**: the pipeline captures most improvements while avoiding unnecessary computation beyond the point of diminishing returns.
>
> > Weaknesses-2: Both the problem proposer and evaluator are GPT-4o-mini, which may create self-judging biases.
>
> **Response:**
>
> We acknowledge the reviewer’s concern regarding potential self-judging bias when both the problem proposer and the evaluator use GPT-4o-mini. While the two agents share the same backbone model, in practice they operate under **different roles, prompts, and objectives**, which already introduces non-trivial distinction between generation and evaluation behaviors. Importantly, the review–revise loop demonstrates clear filtering ability even under this setting:
>
> | Round | Total Inputs | Passed | Pass Rate |
> |-------|--------------|--------|-----------|
> | 1     | 42006        | 7438   | 17.71%    |
> | 2     | 34568        | 6526   | 18.88%    |
> | 3     | 28042        | 4718   | 16.83%    |
> | **All** | — | **18682** | — |
>
> Across iterations, a substantial portion of ambiguous or low-quality problems is removed, while clearer, more coherent, and mathematically valid questions are retained. This indicates that **even with the same underlying model, the evaluator can still effectively distinguish weaker samples from stronger ones**. Furthermore, using *different* models for proposing and evaluating introduces a different type of challenge:  **each model has its own inductive biases**, and mismatched evaluation standards may lead to inconsistent filtering and unstable convergence in multi-round refinement.
>
> Overall, while no evaluation scheme is entirely free of bias, our multi-stage process (review–revise + final quality scoring) consistently improves data quality in a stable manner. We believe this provides a practical and empirically supported balance between evaluation robustness and pipeline consistency.

---

> ### Author Response · Authors · 2025-11-25
> **Response to Reviewer FpEq [2/3]**
>
> > Weaknesses-3.1: The appendix lacks substantial details. It should include more design details (e.g., prompt tuning)
>
> **Response:**
>
> During inference, we fix the sampling temperature to 0 to ensure deterministic outputs, and set the maximum generation length (max tokens) to 2048 across all models. We use a fixed random seed of 0 for reproducibility and set the number of inference trials to 1 for every evaluation. For our primary models, we adopt a standard Chain-of-Thought prompting scheme. Specifically:
> - Training prompt: "Question: {problem}\nAnswer:"
> - Evaluation prompt: "Question: {problem}\nAnswer: Let’s think step by step."
>
> This prompt design follows common practice in math reasoning tasks and ensures that the model is encouraged to generate explicit intermediate reasoning steps. For Mistral 7B and Llama 3 8B, we instead use the Alpaca instruction-following template during inference: "Below is an instruction that describes a task. Write a response that appropriately completes the request: \n\n ### Instruction: \n {problem} \n\n ### Response: \". We chose this template because preliminary experiments showed that Alpaca-style instructions consistently yield better reasoning quality on these two architectures compared with the CoT-style prompt. This observation is also aligned with the findings reported in MathFusion, where Alpaca-style prompting was similarly more effective.
>
> > Weaknesses-3.2: Ablations on the implementation of the “long-tail” diversity selector (e.g., why not use random/clustered selection methods).
>
> **Response:**
>
> To more clearly isolate the contribution of each component in our pipeline, we conducted an ablation study using a 15k synthetic dataset. The full ablation results are shown below:
>
> | Method                     | MATH | GSM8K | College | DM   | Olympiad | Theorem | AVG  |
> |----------------------------|------|-------|---------|------|----------|---------|------|
> | Problem Rephrase           | 28.4 | 76.0  | 22.4    | 35.0 | 8.3      | 18.4    | 31.4 |
> | + Seed Filtering           | 29.2 | 74.1  | 23.7    | 35.1 | 9.6      | 20.4    | 32.0 |
> | + Problem Review–Revise    | 30.5 | 74.2  | 26.0    | 39.8 | 8.7      | 18.5    | 33.0 |
> | + Synthetic Data Evaluation| 31.4 | 74.5  | 25.3    | 40.3 | 8.7      | 18.9    | 33.2 |
>
> Importantly, in the “+ Problem Review–Revise” stage, the 18k refined problems are **randomly sampled** to construct a 15k subset for training. This lets us directly compare random sampling against the **quality-score + long-tail-score ranking** strategy used in “+ Synthetic Data Evaluation.”
>
> The results show that ranking-based selection—prioritizing both high-quality samples and long-tail diversity—consistently outperforms random sampling. This demonstrates that our long-tail selector is not arbitrary, but effectively preserves rare, structurally diverse mathematical patterns that random sampling tends to lose.
>
> The final performance gain of +0.2 after the Synthetic Data Evaluation stage is relatively modest because the preceding multi-round review–revise process has already produced a set of **very high-quality** candidates. With limited noise remaining, the ranking step provides an additional but naturally smaller refinement. Nonetheless, this incremental improvement confirms the value of incorporating long-tail diversity into the final selection stage.
>
> > Weaknesses-3.3: I would also expect an analysis of the pipeline's robustness to agent model choice, e.g., swapping gpt-4o-mini to open models for better reproducibility.
>
> **Response:**
>
> We agree that evaluating robustness under different agent models (e.g., replacing GPT-4o-mini with open-source models) would further strengthen the study. However, running our full 30k-scale pipeline with multiple agent models is computationally expensive. We first filter the 15k seed questions (GSM8K + MATH) and only synthesize data from the retained seed subset. A single end-to-end pipeline run includes:
>
> - **Seed scoring:** 15k evaluations
> - **Rephrase (6×):** ~7k filtered seeds → ~42k generations
> - **Three review–revise rounds:** each round evaluates 20k–40k candidates
> - **Final problem–solution scoring + long-tail ranking:** ~18k evaluations
>
> In total, one full run requires approximately:
>
> - **60k–80k generation calls**, and
> - **50k–70k evaluation calls**,
>
> amounting to **110k–150k model invocations** even for a single agent model.
>
> Repeating this full pipeline for multiple agent models (e.g., Qwen2-Math-7B, Llama-3, DeepSeek-Math) would multiply the cost and require **hundreds of GPU-hours or several hundred USD in API usage**, which is beyond our available resources for this submission.

---

> ### Author Response · Authors · 2025-11-28
> **Response to Reviewer FpEq [3/3]**
>
> [ Continue the response to above Weaknesses-3.3]
>
> That said:
>
> - **The pipeline is model-agnostic**, requiring no model-specific tuning.
> - Users can directly replace GPT-4o-mini with any open-source agent model.
> - We already provide robustness evidence by comparing our approach to MathFusion **under the *same* teacher model (GPT-4o-mini)**, where our pipeline consistently performs better—confirming that gains stem from the **pipeline design**, not merely from model strength.
>
> In summary, while a full robustness sweep over multiple agent models would be valuable future work, the computational overhead makes it infeasible for this revision. Our current experiments still demonstrate that the proposed pipeline is effective and not dependent on a specific model choice.
>
>
> > Question 1: How do you choose the one-shot exemplar in stage 3?
>
> **Response:**
>
> In stage 3, the one-shot exemplar is selected from the model’s own previously generated synthetic examples. Importantly, this exemplar is **not manually crafted**; instead, we **randomly select** one sample from the pool of high-quality, well-formatted outputs produced during earlier synthesis steps. The only requirement is that the example satisfies basic structural criteria—clear reasoning flow, correct formatting, and coherent solution style.
>
> The purpose of providing this one-shot exemplar is not to supply problem-solving hints, but to serve as a **formatting and style anchor**. It helps the model consistently follow the desired reasoning structure and output format, thereby improving stability and reliability in the downstream generation process.
>
> > Question 2: I feel the "problem quality" can be quantified better here. Did you do any analysis of the difficulty distribution, topic distribution, and required skills of the generated data?
>
> **Response:**
>
> To provide additional quantitative evidence of the quality and characteristics of the synthesized data, we conducted a post-hoc analysis using GPT-4o-mini to (i) assign quality scores and (ii) classify each problem into standard mathematical topics.
>
> **1. Quality Score Distribution**
>
> GPT-4o-mini was used as an external evaluator to score all 18,679 refined problems. The resulting distribution is:
>
> | Quality Score | Count | Percentage |
> |---------------|--------|------------|
> | 1             | 307    | 1.64%      |
> | 2             | 1175   | 6.29%      |
> | 3             | 5053   | 27.05%     |
> | 4             | 12142  | 65.00%     |
> | 5             | 2      | 0.01%      |
>
> This indicates that the majority of synthesized questions fall into the high-quality range (score ≥ 4), demonstrating strong clarity, coherent reasoning, and non-trivial complexity.
>
> **2. Topic Distribution of the Final 15k Dataset**
>
> Using GPT-4o-mini as a topic classifier, we analyzed the semantic composition of the final 15k synthetic dataset. The resulting distribution is:
>
> | Topic                     | Count | Percentage |
> |---------------------------|--------|------------|
> | Counting & Probability    | 3705   | 24.70%     |
> | Geometry                  | 3326   | 22.17%     |
> | Algebra                   | 2446   | 16.31%     |
> | Number Theory             | 1475   | 9.83%      |
> | Calculus                  | 1470   | 9.80%      |
> | Precalculus               | 1456   | 9.71%      |
> | Intermediate Algebra      | 907    | 6.05%      |
> | Prealgebra                | 123    | 0.82%      |
> | Linear Algebra            | 71     | 0.47%      |
> | Others                    | 21     | 0.14%      |
>
>
> This shows that the synthesized dataset has broad semantic coverage across major mathematical domains, with strong representation in combinatorics, geometry, algebra, and number theory.
>
> While these statistical analyses are not the core contribution of our work, they demonstrate that the final synthetic dataset produced by our pipeline is both **high-quality** and **diverse in topic coverage**.  Our method’s key strengths—seed filtering, iterative review–revise refinement, and final evaluation—collectively ensure improvements in **clarity, complexity, and structural diversity**, beyond what raw statistics alone can convey.

---

### Official Review · Reviewer_Baih · 2025-11-04

**Soundness:** 2
**Presentation:** 2
**Contribution:** 3
**Rating:** 2
**Confidence:** 4

**Summary:**

This work proposes a novel agentic pipeline for generating high quality synthetic data for math reasoning in LLMs. The pipeline consists of four stages: (1) An existing dataset of seed questions is filtered based on the quality of the questions (2) the seed questions are rephrased into diverse variants (3) the solutions (potentially multiple per question, depending on the desired size) of the new as well as the old questions are rewritten using an LLM (4) the final data is further filtered according to the quality of the problems, retaining only the most high quality ones. The resulting dataset of only 30k-90k questions , when used for finetuning various base models, outperforms (on an average across 6 tasks) baseline synthetic data generation approaches in data-sized matched as well as settings where the baseline approaches have upto 2.3M examples.

**Strengths:**

* The method is simple and results in very sample efficient synthetic data
* The use of DS$^2$ score curation method in the context of synthetic data generation is novel and interesting
* I liked the ablation studies included in Section 4.3

**Weaknesses:**

* Clarity of writing in Section 3.5 could be improved (specifically in Lines 288-291)
* My biggest concern Is that comparisons with MetaMath, Dart-MATH etc. are not fair - because the question generation and more importantly, the solution generation models aren't the same. As evident from the results, most of the gains come from solution augmentation. MetaMath and DART-Math both use weaker teacher models to sample solutions. A fairer comparison would be using GPT-4o-mini as the generation model for both baselines (this can be done on a smaller subset, for eg. 10000 questions, for computational feasibility).
* Multiple important, more recent and stronger baselines seem to be missing from the experimental evaluation. Particularly, Personas based synthetic data generation [1] and  ScaleQuest [2] are relevant baselines (note that similarity of question generation models may not be ensured for ScaleQuest since it trains a special question generator).
* Several experimental details seem to be missing: Number of eval seeds, inference sampling temperature, inference max output length, etc. aren't mentioned

### References.
[1] https://arxiv.org/abs/2406.20094.
[2] https://arxiv.org/abs/2410.18693.

**Questions:**

* Can the authors provide a comparison of the the quality of pure synthetic question with the original set of seed questions (solutions can be re-written bby GPT-4o-mini in both cases)?
* Line 241: How was the revise threshold \tau = 4.5 arrived upon
* What are the statistics of other stages?
* Lines 145-146: What do the authors mean by "eliminating ground truth labels"?
* Line 161 - What do the authors mean by "label-free filtering method?
* On line 89, would  "we evaluate models finetuned on data generated by agentic math….." be more correct? Is AgenticMath the name of the dataset or the proposed pipeline?

### Other Comments.
* Typos in Lines 135-137

---

> ### Author Response · Authors · 2025-11-25
> **Response to Reviewer Baih [1/5]**
>
> Dear reviewer Baih,
>
> We sincerely thank you for the time and effort dedicated to evaluating our submission. We greatly appreciate your thoughtful comments and constructive suggestions, which have helped us identify several aspects where additional clarification or analysis can further strengthen the paper. Your feedback has been invaluable in improving both the rigor and the presentation of our work.
>
> In the following, we provide detailed, point-by-point responses to all weaknesses and questions you raised. We aim to address each concern thoroughly and transparently, and we hope that the additional explanations, experiments, and statistical evidence will clarify the contributions and validity of our approach.
>
> > Weaknesses-1: Clarity of writing in Section 3.5 could be improved (specifically in Lines 288-291)
>
> **Response:**
>
> Thank you for pointing this out. To improve clarity, we have revised the wording in Section 3.5 as follows:
>
> “We first prioritize samples by grouping them according to their quality scores (from 5 down to 0) and selecting high-quality groups first. When the remaining quota falls within a group that contains more samples than needed, we further rank the samples within that group using the long-tail diversity score. This procedure ensures that we always select the highest-quality data available while simultaneously encouraging diversity within groups that exceed the required sampling budget.”
>
> This revised description more clearly explains how quality-based grouping and long-tail ranking interact to guide the final selection process.
>
> > Weaknesses-2: My biggest concern Is that comparisons with MetaMath, Dart-MATH etc. are not fair - because the question generation and more importantly, the solution generation models aren't the same. As evident from the results, most of the gains come from solution augmentation. MetaMath and DART-Math both use weaker teacher models to sample solutions. A fairer comparison would be using GPT-4o-mini as the generation model for both baselines (this can be done on a smaller subset, for eg. 10000 questions, for computational feasibility).
>
> **Response:**
>
> We sincerely appreciate this important observation. Our response is divided into two parts for clarity.
>
> **Part 1 — Fairness Concerns Regarding Teacher Models**
>
> We fully acknowledge the reviewer’s concern about the fairness of comparisons when different teacher models are involved in solution generation. Indeed, prior works such as MetaMath and DART-MATH rely on weaker or inconsistent teacher models, making cross-paper comparisons inherently noisy.
>
> To ensure a more rigorous and fair evaluation, we intentionally eliminate this source of variability by adopting the same teacher model as MathFusion—GPT-4o-mini (2024-07-18)—for generating *all* synthetic data in our pipeline. This design ensures that performance differences arise from the data-generation methodology itself, rather than disparities in teacher model strength.
>
> To further strengthen the fairness discussion, we additionally conduct a 10k-sample controlled comparison across MetaMath, DARTMath, ScaleQuest, and our AgenticMath, all trained under the same SFT setting (Mistral-7B) and evaluated using the **same teacher model (GPT-4o-mini, 2024-07-18)** for solution generation where applicable. The results are shown below:
>
> | Datasets       | Samples | MATH | GSM8K | College | DM   | Olympiad | Theorem | AVG  |
> |----------------|---------|------|--------|---------|------|----------|---------|------|
> | MetaMath       | 10k     | 24.9 | 70.4   | 19.0    | 28.5 | 5.1      | 13.8    | 26.9 |
> | DARTMath       | 10k     | 29.3 | 66.4   | 21.3    | 37.4 | 8.7      | 16.7    | 29.9 |
> | ScaleQuest     | 10k     | 24.9 | 66.4   | 17.5    | 27.3 | 7.7      | 14.1    | 26.3 |
> | **AgenticMath (ours)** | 10k | 29.6 | 73.8 | 24.6 | 38.2 | 7.4 | 16.3 | **31.6** |
>
> This controlled experiment reinforces that our improvements do not stem from stronger teacher supervision—since all methods operate under the **same GPT-4o-mini teacher**—but rather from the **pipeline design itself**.

---

> ### Author Response · Authors · 2025-11-25
> **Response to Reviewer Baih [2/5]**
>
> [Continue the response to above Weaknesses-2]
>
> **Part 2 — “Most Gains Come From Solution Augmentation”**
>
> We appreciate the reviewer’s observation and acknowledge that the presentation of our original experiments may have created the impression that most improvements stem solely from solution augmentation. In reality, the earlier stage “Solution Augmentation” involves not only generating solutions but also doubling the dataset size from 15k to 30k, which naturally contributes to part of the performance increase. This effect was not clearly separated in the initial version and may have caused misunderstanding.
>
> To more precisely isolate the contribution of each component, we conducted an additional ablation study using a fixed 15k synthetic dataset. The results below demonstrate that performance gains accumulate across multiple stages, and are not dominated by solution augmentation alone. Seed filtering, the problem review–revise loop, and synthetic data evaluation all contribute meaningfully to the improvements:
>
> | Method                      | MATH | GSM8K | College | DM   | Olympiad | Theorem | AVG  |
> |-----------------------------|------|-------|---------|------|----------|---------|------|
> | Problem Rephrase            | 28.4 | 76.0  | 22.4    | 35.0 | 8.3      | 18.4    | 31.4 |
> | + Seed Filtering            | 29.2 | 74.1  | 23.7    | 35.1 | 9.6      | 20.4    | 32.0 |
> | + Problem Review–Revise     | 30.5 | 74.2  | 26.0    | 39.8 | 8.7      | 18.5    | 33.0 |
> | + Synthetic Data Evaluation | 31.4 | 74.5  | 25.3    | 40.3 | 8.7      | 18.9    | 33.2 |
>
> > Weaknesses-3: Multiple important, more recent and stronger baselines seem to be missing from the experimental evaluation. Particularly, Personas based synthetic data generation [1] and ScaleQuest [2] are relevant baselines (note that similarity of question generation models may not be ensured for ScaleQuest since it trains a special question generator).
>
> **Response:**
>
> To ensure a fair and controlled comparison of synthetic data quality, our evaluation focuses on baselines that satisfy two key criteria:
> (1) they use **standard SFT training** without additional specialized modules, and
> (2) they **focus specifically on mathematical problem synthesis** and **reference only GSM8K and MATH as seed distributions**, consistent with the setting followed in prior math-data-generation work.
>
> Below we explain why some methods were not included as direct baselines and how we nevertheless incorporate relevant reference results.
>
> **Part 1 — Why Personas-based synthetic data generation [1] is not used as a baseline**
>
> The Personas-based method is not directly comparable because its objective and design differ substantially from math-focused synthesis frameworks. Specifically:
>
> - The method is **not primarily designed for mathematical problem generation**, and its synthesis paradigm does not operate on math seed datasets such as GSM8K or MATH.
> - As noted in its original paper:
>   *“It is also worth noting that the main focus of this work is on creating new synthetic data, unlike much previous research that focuses on generating synthetic outputs for specific inputs (e.g., a math problem).”*
>
> Therefore, while the method is valuable in broader synthetic data research, it does not align with the problem setting of seed-based mathematical data generation. For this reason, including it as a direct baseline would not yield a meaningful or fair comparison in our experimental framework.
>
> **Part 2 — Regarding ScaleQuest [2]**
>
> ScaleQuest trains a specialized question generator. To maintain evaluation fairness, we applied its publicly released ScaleQuest synthetic data to SFT a Mistral-7B model (randomly sampled 30k), and compared it directly with our 30k AgenticMathQA dataset.
>
> The results demonstrate that our synthetic data provides **consistently stronger downstream performance**.
>
> | Methods                           | Samples | MATH | GSM8K | College | DM   | Olympiad | Theorem | AVG  |
> |-----------------------------------|---------|------|--------|---------|------|----------|---------|------|
> | Mistral-7B-ScaleQuest             |  30k    | 32.3 | 78.4   | 27.0    | 40.8 |  8.3     | 19.9    | 34.4 |
> | **AgenticMath-Mistral-7B** | 30k | 35.3 | 79.5   | 27.0    | 41.9 | 11.9     | 19.3    | **35.8** |

---

> ### Author Response · Authors · 2025-11-25
> **Response to Reviewer Baih [3/5]**
>
> > Weaknesses-4: Several experimental details seem to be missing: Number of eval seeds, inference sampling temperature, inference max output length, etc. aren't mentioned
>
> **Response:**
>
> We thank you for pointing out the missing experimental details. We provide the full inference configuration below for clarity.
>
> During inference, we fix the sampling temperature to 0 to ensure deterministic outputs, and set the maximum generation length (max tokens) to 2048 across all models. We use a fixed random seed of 0 for reproducibility and set the number of inference trials to 1 for every evaluation. For our primary models, we adopt a standard Chain-of-Thought prompting scheme. Specifically:
> - Training prompt: "Question: {problem}\nAnswer:"
> - Evaluation prompt: "Question: {problem}\nAnswer: Let’s think step by step."
>
> This prompt design follows common practice in math reasoning tasks and ensures that the model is encouraged to generate explicit intermediate reasoning steps. For Mistral 7B and Llama 3 8B, we instead use the Alpaca instruction-following template during inference: "Below is an instruction that describes a task. Write a response that appropriately completes the request: \n\n ### Instruction: \n {problem} \n\n ### Response: \". We chose this template because preliminary experiments showed that Alpaca-style instructions consistently yield better reasoning quality on these two architectures compared with the CoT-style prompt. This observation is also aligned with the findings reported in MathFusion, where Alpaca-style prompting was similarly more effective.
>
> > Question 1: Can the authors provide a comparison of the the quality of pure synthetic question with the original set of seed questions (solutions can be re-written bby GPT-4o-mini in both cases)?
>
> **Response:**
>
> As shown in the table below, the dataset generated purely by AgenticMat outperforms the original 15k seed dataset in overall average score (33.2 vs. 29.3). More importantly, AgenticMath yields **substantial improvements on multiple out-of-domain benchmarks**, including College, DM, Olympiad, and Theorem.
>
> This demonstrates that the synthetic questions produced by our pipeline not only match—but in many cases exceed—the quality of the seed questions. The stronger performance on out-of-domain tasks indicates that AgenticMath produces problems with **richer structures, more diverse reasoning patterns, and better transferability**. In short, our method does more than replicate the seed distribution: it generates new mathematical problems that generalize more effectively to unseen domains.
>
> | Datasets                     | Samples | MATH | GSM8K | College | DM   | Olympiad | Theorem | AVG  |
> |-----------------------------|---------|------|--------|---------|------|----------|---------|------|
> | Mistral-7B: GSM8K + MATH    | 15k     | 28.6 | 71.1   | 20.3    | 33.4 | 6.8      | 15.8    | 29.3 |
> | AgenticMath-Mistral-7B      | 15k     | 31.4 | 74.5   | 25.3    | 40.3 | 8.7      | 18.9    | 33.2 |
>
> > Question 2: Line 241: How was the revise threshold \tau = 4.5 arrived upon
>
> **Response:**
>
> To further analyze the effect of the revise threshold τ_rev, we performed an additional sensitivity study with τ_rev ∈ {3.5, 4.0, 4.5} using Llama3-8B:
>
> | Revise Threshold | Samples | MATH | GSM8K | College | DM   | Olympiad | Theorem | AVG  |
> |------------------|---------|------|--------|---------|------|----------|---------|------|
> | 4.5              | 30k     | 36.8 | 78.4   | 29.6    | 40.3 | 11.4     | 20.4    | 36.2 |
> | 4.0              | 30k     | 36.6 | 77.5   | 28.2    | 43.1 | 11.5     | 20.0    | 36.2 |
> | 3.5              | 30k     | 37.8 | 77.4   | 27.4    | 41.0 | 10.3     | 20.0    | 35.7 |
>
> We observe that **τ_rev = 4.0 and 4.5 yield highly consistent performance**, indicating that our review–revise mechanism is robust across reasonable settings. In contrast, **τ_rev = 3.5 performs worse**, which is expected because a looser threshold allows more low-quality candidates to pass into later stages, weakening the overall dataset quality.

---

> ### Author Response · Authors · 2025-11-26
> **Response to Reviewer Baih [4/5]**
>
> > Question 3: What are the statistics of other stages?
>
> **Response:**
>
> To provide a clearer understanding of the robustness and transparency of our data generation pipeline, we present detailed statistics for all major stages, including seed scoring, rephrase expansion, the multi-round review–revise refinement process, and the final data evaluation. These analyses highlight how our pipeline systematically improves **problem diversity, clarity, and complexity**, which are essential properties for enhancing model reasoning ability.
>
> **1. Seed Data Scoring**
>
> We first assess all raw seed questions using our label-free scoring mechanism. A total of 7,001 questions meet the filtering threshold (score ≥ 3). The full distribution is provided below:
>
> | Dataset | Score 0 | Score 1 | Score 2 | Score 3 | Score 4 | Score 5 | Filtering (score ≥ 3) |
> |---------|---------|---------|---------|---------|---------|---------|------------------------|
> | GSM8K   | 823     | 3522    | 1849    | 1188    | 91      | 0       | 1279                   |
> | MATH    | 69      | 825     | 884     | 2053    | 3415    | 254     | 5722                   |
>
> This stage ensures that only seed questions with sufficient structural soundness and *baseline complexity* are used for further synthesis.
>
> **2. Rephrase Expansion**
>
> To enhance problem **complexity** and **diversity** while preserving essential semantics, each filtered seed question is rephrased six times. This expands the problem space to:
>
> | Stage       | Count Calculation          | Total |
> |-------------|-----------------------------|--------|
> | Rephrase | (1279 + 5722) × 6           | 42006  |
>
> A total of 42,006 synthesized candidates are passed to the review–revise process.
>
> **3. Review–Revise Loop**
>
> Our three-round review–revise framework progressively improves **clarity**, **logical correctness**, and **Mathematical Validity** of the synthesized problems. Detailed statistics are as follows:
>
> | Round | Total Inputs | Passed | Pass Rate |
> |-------|--------------|--------|-----------|
> | 1     | 42006        | 7438   | 17.71%    |
> | 2     | 34568        | 6526   | 18.88%    |
> | 3     | 28042        | 4718   | 16.83%    |
> | All | — | 18682 | — |
>
> Across rounds, lower-quality or ambiguous problems are removed, while more well-structured and coherent questions are retained.
>
> **4. Synthetic Data Quality Distribution**
>
> After the full refinement pipeline, 18,679 high-quality synthetic problems remain. Their scoring distribution is:
>
> | Score | Count | Percentage |
> |-------|--------|-------------|
> | 1     | 307    | 1.64%       |
> | 2     | 1175   | 6.29%       |
> | 3     | 5053   | 27.05%      |
> | 4     | 12142  | 65.00%      |
> | 5     | 2      | 0.01%       |
>
> The fact that 65% of the data is rated at score 4, with an additional 27% at score 3, shows that the majority of synthesized questions exhibit **high clarity**, **substantial reasoning depth**, and **meaningful problem complexity**—properties that are essential for improving downstream mathematical reasoning.
>
> To construct the final dataset used in our experiments, we further rank all refined samples using a combination of the quality score and the long-tail score, prioritizing both overall quality and distributional diversity. From this ranked list, we select the top **15k** problems as the final synthetic dataset for training.
>
> Overall, these statistics demonstrate that our pipeline is not only scalable and reliable, but also highly effective at producing **diverse, clear, and complex** mathematical problems, enabling substantial improvements in model performance.
>
> > Question 4 : Lines 145-146: What do the authors mean by "eliminating ground truth labels"?
>
> **Response:**
>
> In our method, “eliminating ground truth labels” means that our synthetic data generation pipeline does not rely on any reference solutions from the seed dataset. We only use the seed questions as input, and the model autonomously generates both the revised problems and the corresponding solutions. In other words, our pipeline does not require access to the original seed answers during synthesis. This design removes the dependency on ground-truth labels and allows the method to operate even when only unlabeled problem statements are available.

---

> ### Author Response · Authors · 2025-12-02
> **Response to Reviewer Baih [5/5]**
>
> > Question 5: Line 161 - What do the authors mean by "label-free filtering method?
>
> **Response:**
>
> Compared with DART-MATH, which assesses problem difficulty by measuring model accuracy against ground-truth answers, our method does not rely on any ground-truth labels. DART-MATH requires the correct solution for each question in order to evaluate whether the model's prediction is correct, and this correctness signal is then used to classify difficulty levels. In contrast, our filtering approach evaluates the quality of a question based solely on the question text itself, without using or referencing any ground-truth solution. This design is particularly important for synthetic problems, which do not come with ground-truth labels. Therefore, our method remains applicable even in fully label-free settings.
>
> > Question 6: On line 89, would "we evaluate models finetuned on data generated by agentic math….." be more correct? Is AgenticMath the name of the dataset or the proposed pipeline?
>
> **Response:**
>
> Thank you for the clarification request. To avoid confusion: AgenticMath refers to our proposed data generation pipeline, whereas AgenticMathQA refers to the resulting synthetic dataset produced by this pipeline. Therefore, the correct phrasing on line 89 should indeed be “we evaluate models finetuned on data generated by AgenticMath,” since the term refers to the pipeline rather than the dataset itself.

---

### Note · Authors · 2026-01-05

I have read and agree with the venue's withdrawal policy on behalf of myself and my co-authors.